# Synthesizing Images on Perceptual Boundaries of ANNs for Uncovering and Manipulating Human Perceptual Variability

## Abstract

Human decision-making in cognitive tasks and daily life exhibits considerable variability, shaped by factors such as task difficulty, individual preferences, and personal experiences. Understanding this variability across individuals is essential for uncovering the perceptual and decision-making mechanisms that humans rely on when faced with uncertainty and ambiguity. In this study, we present a *counterfactual-based approach* to investigate the subject-level decision-making behaviors and reveal the underlying perceptual mechanisms by synthesizing visual stimuli. First, we developed an efficient generative model that samples along an artificial neural network (ANN)'s perceptual boundary, generating image samples designed to induce high variability in human perception. Using these generated samples, combined with behavioral data from 246 human participants across 116,715 trials, we constructed the varMNIST dataset. Then, we presented a subject-specific fine-tuning approach to align the perceptual variability of ANNs with that of humans. It allows us to successfully predict human decision-making behaviors on varMNIST. Finally, we verified the ability to selectively manipulate individual behaviors by generating tailored controversial stimuli, which highlighted significant inter-subject perceptual variability. Together, our work illuminated key distinctions between human and machine perceptual variability and established an effective strategy for manipulating individual decision-making behaviors. This study paves the way for artificial intelligence models with personalized perceptual capabilities.

## 1 Introduction

A core goal of cognitive science is to establish models that reflect the relationship between external stimuli and human internal experiences. The development of ANNs has significantly contributed to this goal, particularly through the latent representations of ANNs that have shown a strong correlation with human psychological representations (Wei et al. (2024a;b); Muttenthaler et al. (2022a); Mahner et al. (2024); Zheng et al. (2019); Hebart et al. (2020); Muttenthaler et al. (2022b)). However, individuals may have markedly different internal experiences when presented with the same external stimulus (Snyder et al. (2015); Partos et al. (2016); Floridou et al. (2022). For instance, when different people observe the same handwritten digit, they may recognize it as different numbers (see Figure 1 (left)). This *variability* in human percepts has been inadequately explored within both cognitive science and computer science. We hypothesize that there is a correspondence between the perceptual boundaries of ANNs and human perceptual classification boundaries and images generated along these boundaries would evoke different internal experiences in humans. This hypothesis can be tested by human experiments with visual stimuli sampled along the perceptual boundaries of ANNs.

In recent years, counterfactual generation methods have made significant progress in revealing the differences between human and machine perception. Veerabadran et al. (2023); Zhou & Firestone (2019); Elsayed et al. (2018) highlighted that while ANNs exhibit brittleness in response to small perturbations, these perturbations can also bias human perceptual choices under specific conditions. Similarly, Gaziv et al. (2024) found that by enhancing ANN models, low-norm image perturbations could be generated that significantly disrupt human percepts, demonstrating their potential as pre-

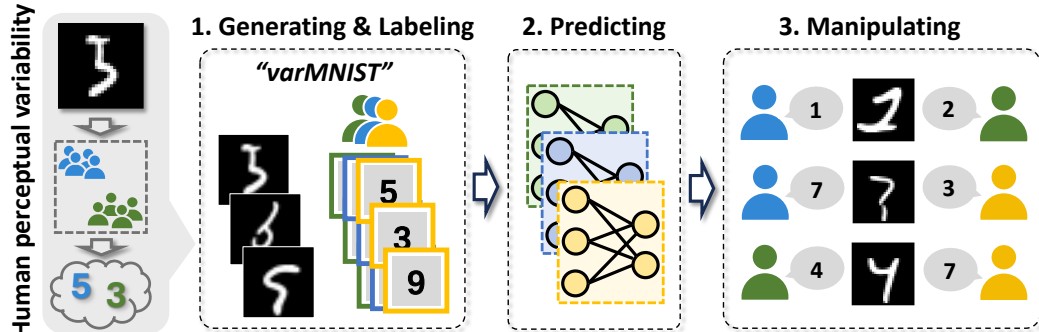

Figure 1: **Overview of our paradigm.** Our approach consists of three main components: **1. Generating & labeling:** Sampling images from ANN decision boundaries and using them in human behavioral experiments to construct the high-variability dataset *varMNIST*; **2. Predicting:** Fine-tuning models with human behavioral data to align them with human perceptual variability at the group and individual levels, enhancing behavior prediction accuracy; **3. Manipulating:** Employing individually fine-tuned models to generate images that elicit high perceptual differences between individuals, with the manipulations validated through human experiments.

cise tools for altering human category perception. However, Feather et al. (2023; 2019) discovered that existing neural networks often produce "model metamers" that are unrecognizable to humans, further emphasizing the misalignment between models and human perception. Additionally, Golan et al. (2020; 2023) introduced the concept of "controversial stimuli" to compare different models, revealing not only discrepancies among models but also substantial divergences from human perception in vision and language. Together, these studies suggest that counterfactually generated visual stimuli can offer valuable insights into the variability of human percepts.

In this study, we present a paradigm for studying *human perceptual variability* through image generation. As shown in Figure 1, our approach includes three main components: **1. Generating & labeling.** We sample images from the decision boundaries of ANNs and use these synthetic images in human behavioral experiments to construct a high-variability dataset, *varMNIST*, which captures perceptual variability across individuals. **2. Predicting.** Human behavioral data are used to finetune models, aligning them with human perceptual variability at both the group level and individual level. This allows us to evaluate how well neural networks can predict human perception. **3. Manipulating.** Individual finetuned models are leveraged as adversarial classifiers to generate new images that amplify perceptual differences between individuals. These images are validated through human experiments to assess their ability to enhance and decode individual perceptual variability. This paradigm provides a systematic framework for investigating and manipulating human perceptual variability, bridging the gap between neural networks and human perception.

We have three main contributions.

(1) **A novel generative approach to probe human perception**: We developed an efficient generative method that samples along ANN perceptual boundaries to generate naturalistic images, forming the varMNIST dataset (Figures 2, A.1). Our human experiments demonstrated that the varMNIST dataset successfully evoked high variability in human perception (Figure 3).

(2) **Aligning human and ANN perceptual variability**: Through subject-specific fine-tuning, we aligned the perceptual variability of ANNs with that of humans, enabling better predictions of human decision-making behaviors (Figure 4).

(3) **Revealing and manipulating individual decisions**: We verified that controversial stimuli can selectively manipulate individual decision-making behaviors by human experiments. These stimuli unveil significant inter-individual differences in perceptual variability (Figure A.16) and successfully manipulate individual behaviors (Figures 5, A.17, A.18, A.19).

## 2    RELATED WORKS

Researchers have extensively used synthetic images generated by ANNs to study human perceptual space, uncovering differences between model and human perception while refining generation techniques to enhance their influence on human cognition. For instance, Golan et al. (2020; 2023) utilized controversial stimuli to highlight classification discrepancies between neural networks. Similarly, Veerabadran et al. (2023) demonstrated that adversarial perturbations could simultaneously influence ANN classifications and human perceptual choices, revealing shared sensitivities. However, Gaziv et al. (2024) found that while standard ANN perturbations fail to impact human perception, robustified ANN models can generate low-norm perturbations that significantly disrupt human percepts.

Other studies have approached this problem from different angles. For example, works like Feather et al. (2023; 2019); Nanda et al. (2022; 2023) investigated *model metamers*, revealing fundamental mismatches between model activations and human recognition. Extending beyond perceptual discrepancies, Fu et al. (2023) introduced DreamSim, a perceptual metric leveraging synthetic data and human experimental data to better reflect human similarity judgments and address shortcomings in conventional perceptual metrics. Building on such synthetic data and behavioral insights, recent efforts have sought to align vision models with human perceptual representations by incorporating human-like conceptual structures, resulting in improved alignment and enhanced performance across diverse tasks Muttenthaler et al. (2024); Sundaram et al. (2024).

To study the variability of human perception, it is essential that generated images significantly influence human cognition. Given that we sample from the perceptual boundaries of ANNs, which often contain high noise levels, better methods are needed to ensure that the generated images appear natural. Recently, the fields of adversarial examples and counterfactual explanations in machine learning have adopted effective techniques to help deal with this problem, such as Jeanneret et al. (2023), Wei et al. (2024b), Chen et al. (2023), Jeanneret et al. (2022), Vaeth et al. (2023), and Atakan Bedel & Çukur (2023). These studies use diffusion models with training-free guidanceYu et al. (2023); Ma et al. (2023); Yang et al. (2024) as regularizers to introduce prior distributions, thereby enhancing the naturalness of generated images and their impact on human perception.

## 3    COLLECTING HUMAN PERCEPTUAL VARIABILITY

In this section, we introduce the method for constructing varMNIST: a digit dataset with high perceptual variability. Our goal is to generate images that evoke significant human perceptual variability and collect this variability by recording human perceptual judgments on the generated images.

### 3.1    GENERATING IMAGES BY SAMPLING FROM THE PERCEPTUAL BOUNDARY OF ANNS

The image perturbations that significantly affect ANN perception also influence human perception ( Gaziv et al. (2024); Veerabadran et al. (2023); Wei et al. (2024a); Muttenthaler et al. (2022a)), suggesting that ANNs and humans may share similar perceptual boundaries. Based on this, we hypothesize that samples on these boundaries (which exhibit high perceptual variability for ANNs) may also lead to ambiguous perception in humans, resulting in different internal experiences for the same stimuli.

We adopted two guidance strategies: uncertainty guidance and controversial guidance. Uncertainty guidance aims to generate images that lie near the decision boundaries of classifiers. Its loss function is defined as:

$$\mathcal{L} = H(p_1(y|x), q_1(y))$$

where $H(p,q)$ is the cross-entropy function that measures the discrepancy between the predicted distribution $p(y)$ and the target distribution $q(y)$. The target distribution ensures equal probabilities for two categories (e.g., "3" and "5"), resulting in high-uncertainty images. Controversial guidance generates images that cause conflicting predictions between two classifiers. Its loss function is defined as:

$$\mathcal{L} = H(p_1(y|x), q_1(y)) + H(p_2(y|x), q_2(y)),$$

where $p_1(y|x)$ and $p_2(y|x)$ are the predicted probability distributions of classifiers 1 and 2, and $q_1(y)$ and $q_2(y)$ are their corresponding target distributions. The target distributions ensure that

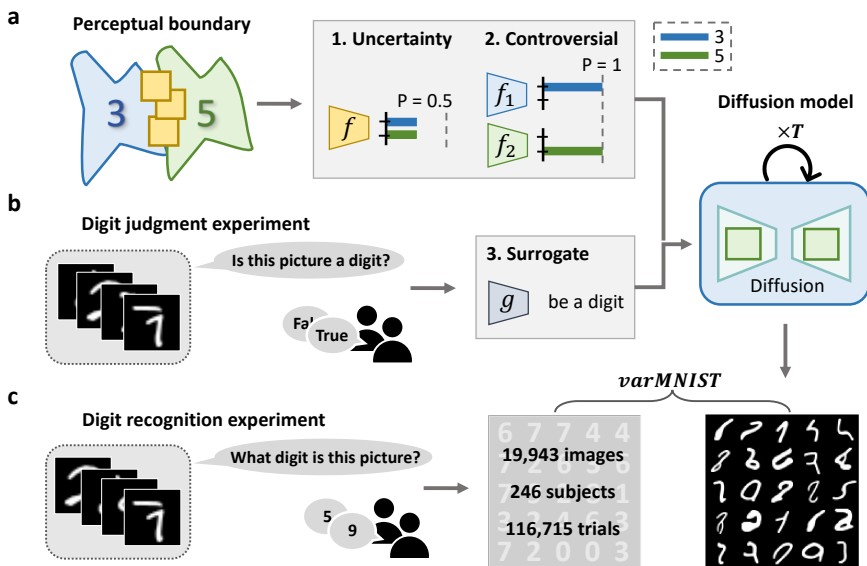

Figure 2: **Generating images to elicit human perceptual variability.** (a) The example illustrates two guidance methods for sampling from the perceptual boundary between "3" and "5" in ANN: *uncertainty guidance* and *controversial guidance*. Specifically, *Uncertainty guidance* aims to make the ANN model $f$ assign equal probabilities to "3" and "5," while *controversial guidance* generates images classified as "3" by $f_1$ but as "5" by $f_2$. One of these guidance methods is incorporated into the image generation process. (b) The synthetic images were used in a digit judgment experiment where participants answered, "Is this picture a digit?" We trained a *digit judgment surrogate* based on human responses and used it as a classifier to guide the image generation process. (c) We used the images synthesized using the two guidance methods, ANN perceptual boundary sampling and digit judgment surrogate, for the digit recognition human experiment. Participants were asked "What digit is this picture?" A total of 19,952 images were used, with 123,000 trials conducted across 246 participants, resulting in the high perceptual variability dataset varMNIST.

classifier 1 predicts one category (e.g., "3") with high confidence, while classifier 2 predicts another category (e.g., "5") with high confidence, generating controversial images. Figure A.1 illustrates the guidance methods. Details of additional analyses and comparisons of guidance methods can be found in Appendix A.2.

Previous studies have shown that when using generated images to investigate models and human perception (e.g., Golan et al. (2020); Gaziv et al. (2024); Veerabadran et al. (2023); Feather et al. (2023)), a common issue is the lack of naturalness in the generated images. This often makes the images difficult for participants to recognize, thereby weakening their impact on human cognition (see Figure A.2). Recent research has demonstrated that diffusion models, when used as regularizers, can introduce prior information and help generate more natural images Jeanneret et al. (2023); Wei et al. (2024b); Chen et al. (2023); Jeanneret et al. (2022); Vaeth et al. (2023); Atakan Bedel & Çukur (2023). Building on these findings, we employ a classifier-guided diffusion model for image generation. This method produces images that are closer to the true distribution of handwritten digits, thereby significantly enhancing their impact on human perception (see Appendix A.1).

### 3.2 IMPROVING GENERATIVE QUALITY BY DIGIT JUDGMENT SURROGATE

**Digit judgment experiment.** We used the synthetic images as experimental stimuli to measure human behavior in a digit judgment task. The purpose of this experiment was to collect human judgments on whether a given image qualifies as a digit, thereby establishing a human criterion for handwritten digit. For each image, participants were asked the question, "Is this image a digit?" with responses limited to "True" or "False." More experimental details can be found in Appendix A.4.1.

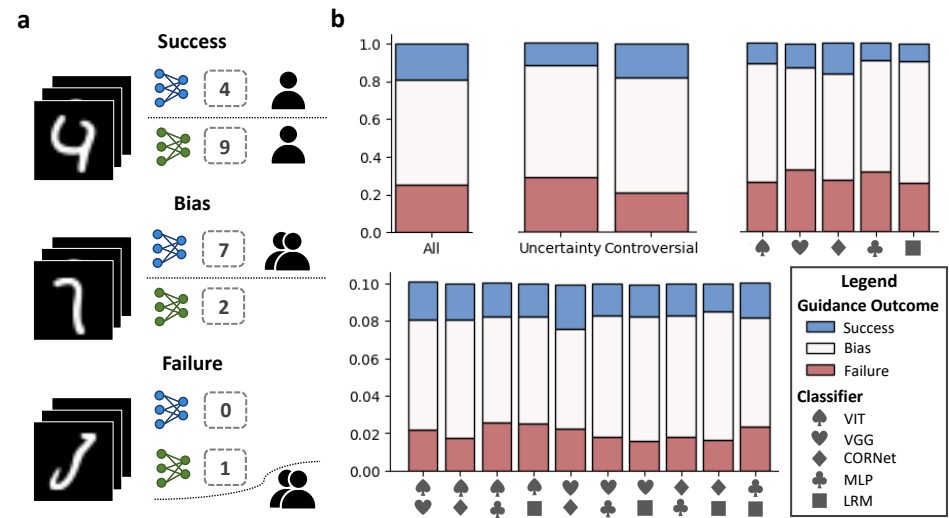

Figure 3: **Quantitative analysis of varMNIST.** (a) Examples of three types of *guidance outcome*: *success*, *bias*, and *failure*. (b) Guidance outcomes across strategies and classifiers. The average sum of overall *success* and *bias* rates approaches 80%. Controversial guidance achieves a higher *success* rate than uncertainty guidance, with similar *bias* rates. CORNet performs best in uncertainty guidance, while LRM performs worst. In controversial guidance, combinations of VGG and CORNet achieve the highest *success* rates and lowest *bias* rates, but exhibit relatively high *bias* rates when paired with other classifiers.

The experiment collected behavioral data from 400 participants, comprising 200,000 trials and 20,000 stimuli. During the data cleaning process, 124 participants were excluded based on Sentinel trials, leaving data from 276 participants (138,000 trials and 19,878 valid stimuli). This dataset provides a robust foundation for analyzing perceptual standards for handwritten digits.

**Guiding generative process by digit judgment surrogate.** For any given image, we use the frequency of participants responding "True" as the probability of the image being a digit. The initial image dataset, along with the corresponding probabilities, was used to train a digit judgment surrogate. As the previous works of image generation by human preferences (Liang et al. (2024); Bansal et al. (2023)), this surrogate, functioning as a image quality predictor, was then employed to guide the image generation process (see Appendix A.3.3). The guidance formula can be expressed as:

$$\mathcal{L}_{total} = \mathcal{L} + max((1 - f_{surr}(x))^2, 0.5)$$

In this formula, $\mathcal{L}_{total}$ represents the total loss. $f_{surr}(x)$ represents the probability give by the digit judge model. The probability of the digit judge is combined to the formula to ensure the generated image is considered as a digit by humans. The max function is used so that when the score is above a certain threshold, the gradient of the digit judge will not effect generation.

### 3.3 MEASURING HUMAN PERCEPTUAL VARIABILITY BY RECOGNITION EXPERIMENT

#### 3.3.1 DIGIT RECOGNITION EXPERIMENT.

We used the image dataset generated through uncertainty or controversial guidance and digit judgment surrogate guidance as experimental samples to measure human behavior in a digit recognition task. For each test image, participants were asked, "What number is this image?" with responses restricted to one of the digits from 0 to 9. We collected the probability distributions of human responses and calculated the average response time and entropy distribution for all test images (Figure A.10). The experiment collected behavioral data from 400 participants, each completing 500 trials, resulting in a total of 200,000 trials across 20,000 stimuli. During data preprocessing, 154

participants were excluded based on Sentinel trials, leaving data from 246 participants (116,715 trials and 19,943 valid stimuli). Using this cleaned dataset, we constructed a high perceptual variability dataset, varMNIST, which serves as a foundation for subsequent analysis and modeling.

### 3.3.2 QUANTITATIVE ANALYSIS OF VARMNIST

**Evaluation metrics.** To comprehensively evaluate the guiding effectiveness of the generation method, we define three types of *guidance outcome*, as illustrated in Figure 3a: *success*, *bias*, and *failure*. For the guidance targets $o_1$ and $o_2$, let $p_1$ and $p_2$ represent the probabilities of participants choosing $o_1$ and $o_2$, respectively. A result is considered *success* if $p_1 + p_2 \geq 80\%$ and $\min(p_1, p_2) \geq 10\%$, indicating the generated stimuli guide participants to make a balanced choice between the two targets. A result is labeled as *bias* if $p_1 + p_2 \geq 80\%$ but $\min(p_1, p_2) < 10\%$, indicating a strong bias toward one target. A result is classified as *failure* if $p_1 + p_2 < 80\%$, meaning the stimuli fail to guide participants effectively. These definitions allow us to evaluate and compare the performance of different guidance strategies and classifiers.

**ANN variability can arouse human variability.** To evaluate whether the images generated by sampling on the perceptual boundaries of ANNs can arouse human perceptual variability, we first calculated the entropy of participants' choice probabilities in the digit recognition experiment. As shown in Figure. A.10 (bottom left), the entropy values for more than half of the generated images were significantly greater than zero, indicating substantial variability in human choices. This suggests that the generated images successfully elicited human perceptual variability. Furthermore, as illustrated in Figure. 3b, the average sum of success rate and bias rate across all generated images was close to 80%. This indicates that, in the majority of cases, human choices aligned with either both or one of the guidance targets. This demonstrates that the generation method effectively guided human digit recognition behavior.

**Guidance strategy, classifier and target influence the guidance outcome.** We further analyzed the influence of different guidance strategies, classifiers used for guidance, and guidance targets on the guidance outcome. First, for the two guidance strategies, uncertainty guidance and controversial guidance, the success rate of controversial guidance is higher, while the bias rates of both methods are similar (Figure. 3b). This indicates that controversial guidance is more effective in guiding participants. Second, for the five classifiers used in uncertainty guidance, CORNet achieved the highest success rate and the lowest failure rate, indicating that CORNet is the most effective guidance classifier (Figure. 3b). This result aligns with the fact that CORNet incorporates more human visual priors. For the ten adversarial classifier combinations used in controversial guidance, the combination of VGG and CORNet achieved the highest success rate and the lowest bias rate, demonstrating that VGG and CORNet have strong guiding capabilities, with comparable guidance strength. Additionally, when VGG and CORNet were combined with other classifiers, their bias rates were relatively high, with the combination with LRM exhibiting the highest bias rate. This suggests that other classifiers are less effective compared to VGG and CORNet, with LRM being the least effective in guidance strength. Finally, for the ten guidance targets (digits 0-9), the results showed significant variability in guidance outcomes (Figure. A.12). For example, digit pairs (1, 7), (1, 2), and (4, 9) achieved the highest success rates, exceeding 0.35. In contrast, digit pairs (1, 8), (2, 9), and (7, 8) exhibited the lowest success rates, below 0.03. These findings highlight that the guidance outcome is strongly influenced by the specific digit pairs being guided.

## 4 PREDICTING HUMAN PERCEPTUAL VARIABILITY

### 4.1 MODEL FINE-TUNING FOR HUMAN ALIGNMENT

To align models with both group-level and individual-level performance, we adopted a mixed training approach with an 80:20 split for training and validation. For individual-level datasets (varMNIST-i), the validation set was designed to avoid overlap with the group validation set. For group-level training, we combined the MNIST and varMNIST datasets in a 1:1 ratio, ensuring performance on MNIST while fine-tuning for perceptual variability. For individual-level training, we mixed varMNIST-i, varMNIST, and MNIST datasets in a 2:1:1 ratio, ensuring the models performed effectively on individual-specific, group, and original datasets. See Appendix B.2 for more details.

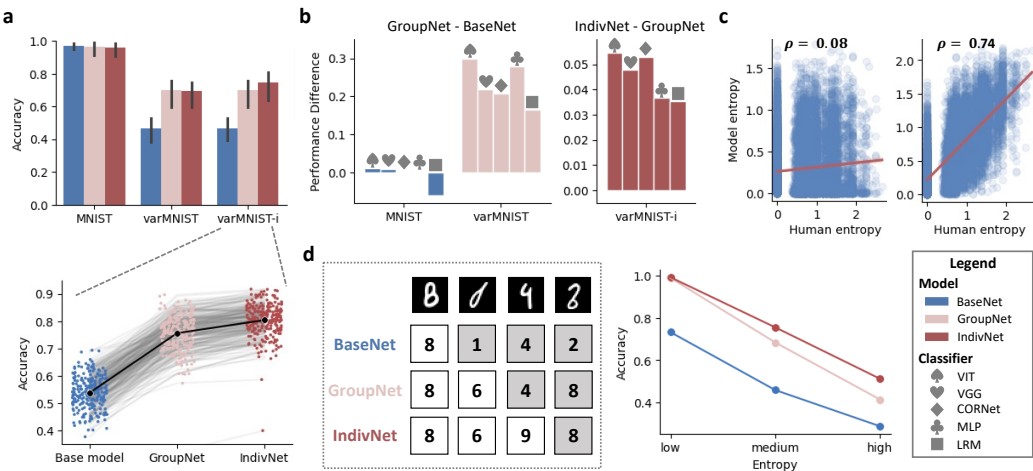

Figure 4: **Human alignment results.** (a) Accuracy of BaseNet, GroupNet, and IndivNet on MNIST, varMNIST, and varMNIST-i. All models performed similarly on MNIST. On varMNIST, GroupNet and IndivNet improved accuracy by ~20% over BaseNet, with IndivNet outperforming GroupNet by 5% on varMNIST-i. Accuracy improved for 241 participants and decreased for 5 after inividual fine-tuning. (b) Fine-tuning results for five classifiers. On MNIST, group fine-tuning improved VIT and VGG, while others remained unchanged or declined. On varMNIST, all classifiers improved, with VIT and MLP showing the largest gains and LRM the smallest. Individual fine-tuning further improved all classifiers with the same trend. (c) For VGG, Spearman rank correlation between model and human entropy increased from $\rho = 0.08$ to $\rho = 0.74$ after group fine-tuning. (d) Performance of BaseNet, GroupNet, and IndivNet of varying entropy levels. The choices from selected subject for the example images are 8, 6, 9, 6, with increasing entropy levels. GroupNet and IndivNet improved over BaseNet on all entropy levels, while IndivNet's gains over GroupNet were focused on high-entropy images.

## 4.2 ALIGNMENT ANALYSIS ON VALIDATION DATASETS

**Fine-tuning improves both group-level and individual-level prediction performance.** As shown in Figure 4a, BaseNet, GroupNet, and IndivNet achieve nearly identical prediction accuracy on the MNIST dataset, indicating no significant loss of baseline performance after fine-tuning. On the varMNIST dataset, both GroupNet and IndivNet outperform BaseNet by approximately 20%. Furthermore, IndivNet achieves an additional ~5% accuracy improvement over GroupNet on the varMNIST-i dataset, demonstrating its superior adaptability to individual differences. After individual fine-tuning, accuracy improved for 241 participants, while only 5 participants experienced a slight decrease, highlighting the effectiveness of individual fine-tuning in adapting to unique participant behavior and capturing human perceptual variability more accurately.

**Different classifiers exhibit inconsistent performance.** Figure 4b and A.13 compares the fine-tuning performance of five classifiers. On the MNIST dataset, group fine-tuning improved the prediction accuracy of VIT and VGG, while CORNet and MLP remained unchanged, and LRM showed a significant decrease in accuracy. On the varMNIST dataset, all classifiers exhibited improvements, with VIT and MLP achieving the largest gains and LRM the smallest. Individual fine-tuning further improved all classifiers, with VIT and MLP showing the greatest adaptability to fine-tuning, while LRM demonstrated weaker generalization ability. These results highlight that both group- and individual-level fine-tuning can significantly enhance classifier performance, but the degree of improvement depends on the classifier architecture.

**Human variability can be predicted by models.** To evaluate the alignment between model and human perceptual variability, we analyzed the correlation between model and human entropy, as shown in Figure 4c and A.14. Taking VGG as an example, group fine-tuning increased the Spearman rank correlation between model and human entropy from $\rho = 0.08$ to $\rho = 0.74$. This signifi-

cant improvement indicates that fine-tuning enables the model to better capture human uncertainty, aligning model predictions more closely with human perceptual behavior.

**Performance of behavior prediction across images with varying entropy levels.** Image entropy reflects task difficulty, with higher entropy indicating more challenging samples. To examine the impact of entropy levels on prediction accuracy, we analyzed model performance across varying entropy levels, as shown in Figure 4d and A.15. Both GroupNet and IndivNet outperform BaseNet across all entropy levels, demonstrating that fine-tuning enhances prediction accuracy regardless of task difficulty. Notably, IndivNet's performance gains over GroupNet are most pronounced for high-entropy images, suggesting that individual fine-tuning primarily improves prediction accuracy for difficult samples. These findings highlight the ability of fine-tuned models to better handle challenging stimuli, capturing subtle variations in human perceptual behavior more effectively.

## 5 Manipulating human perceptual variablity

### 5.1 Experimental paradigm

Building on varMNIST and alignment experiments, we designed a paradigm to test whether individually fine-tuned models can amplify perceptual differences and guide decision-making (Figure 5a). This experiment evaluates the ability of targeted stimuli to reveal individual variability and achieve precise manipulation of perceptual outcomes, highlighting the potential of personalized modeling in understanding human perception. For the *first round* of experiments, we initially selected around 500 balanced samples from the varMNIST dataset as stimuli. After collecting behavioral data from pairs of participants, we fine-tuned their individual models using the method described in Section 4.1. Controversial stimuli were then generated using the updated models as described in Section 3, aiming to elicit distinct choices between the two participants, with each choosing their respective guidance targets.

In the *second round* of experiments, these controversial stimuli were presented to participants in pairs, with each pair completing trials designed to test whether the fine-tuned models could effectively guide their decisions in opposite directions. The goal was to evaluate whether the generated stimuli amplified perceptual differences and aligned participants' responses with their respective guidance targets. For each subject pair, approximately 180 controversial samples were generated, ensuring the sample distribution remained as balanced as possible. A total of 18 participants were recruited for in-lab experiments, grouped into six sets of three participants each. Within each group, participants were paired in all possible combinations, resulting in three pairs per group and 18 pairs overall. Each participant completed 500 trials in the first round and approximately 360 trials (180 per pair, across two pairs) in the second round.

### 5.2 Manipulating Results

**Evaluation metrics.** To analyze the effects of individual manipulation, we employed two key metrics. The first metric, referred to as the **guidance outcome** (Figure 5b), was adapted from Section 3.3.2. It categorizes outcomes for two participants, $s_1$ and $s_2$, with respective guidance targets $o_1$ and $o_2$, and choices $c_1$ and $c_2$. A result is labeled as *success* if both participants' choices fall within their respective guidance targets and are distinct, i.e., $c_1, c_2 \in \{o_1, o_2\}$ and $c_1 \neq c_2$. If both choices are biased toward the same target, such as $c_1 = c_2 = o_1$ or $o_2$, it is categorized as *bias*. Finally, if at least one choice is outside the targets ($c_1, c_2 \notin \{o_1, o_2\}$), the outcome is labeled as *failure*. The second metric, called the **targeted ratio** (Figure 5c), quantifies the directionality of successful guidance. Within successful trials, participant choices are classified as either *positive*, where $c_1 = o_1$ and $c_2 = o_2$, meaning both choices align with their respective targets, or *negative*, where $c_1 = o_2$ and $c_2 = o_1$, indicating swapped choices. The targeted ratio is defined as the proportion of positive trials among all success trials, providing a measure of the effectiveness of directional guidance. We present examples of stimuli demonstrating various guidance outcomes and directions in Figure A.20.

**Improvement in guidance outcome.** We first analyzed the improvements in the guidance outcome achieved through individual manipulation. As shown in Figure. 5, A.17, A.18, A.19, com-

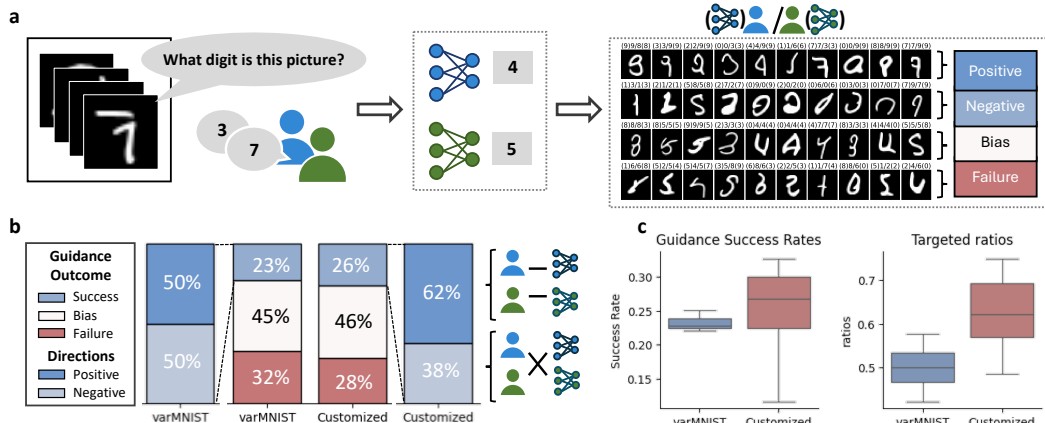

Figure 5: **Manipulation analysis.** (a) Individual manipulation experiment process. Participants first completed the *first round*, and these behavioral data were used to finetune their individual models. Based on the finetuned models, pairwise targeted controversial guidance was used to generate stimuli, which were used in the *second round*. The left panel showcases examples of success (positive, negative), bias, and failure cases, illustrating the choices made by individual models and human participants. (b) The middle two bars show the guidance outcomes for varMNIST and the individually customized dataset, with the latter achieving a higher success rate. The left and right bars further analyze the successful samples, where the dark blue indicates the participant's choices aligned with the guidance direction, and the light blue indicates the opposite. Compared to varMNIST, IndivNets also improves the directionality of guiding perceptual changes. (c) The left panel shows the guidance success rates for the first-round stimuli and the second-round stimuli generated by the finetuned models, with an improvement of ∼3% ($p < 0.001$). The right panel shows the *targeted ratios* (i.e., the proportion of participant choices aligned with the guidance direction) for these two groups of stimuli, with an increase of ∼12% ($p < 0.001$).

pared to varMNIST, the success rate in the individually customized dataset increased by 3%, the bias rate increased by 1%, and the failure rate decreased by 4%. Considering that each participant completed only around 200 samples in the experiment, compared to 20,000 samples in varMNIST, this represents a very small sample size. Therefore, these results indicate that even with a limited sample size, we successfully captured the perceptual differences among individual participants. These findings validate the feasibility and effectiveness of individual manipulation using small, customized datasets, demonstrating that precise modeling and manipulation of human perceptual behavior can be achieved even at low cost.

**Improvement in guiding directionality.** We further evaluated the guiding directionality in successful trials. As shown in Figure. 5, A.17, A.18, A.19,, compared to varMNIST, the target ratio of IndivNets improved by 12%, indicating a significant enhancement in the directional guidance achieved with individually customized datasets. This result suggests that individual fine-tuning not only improves the model's guiding capability but also enables more precise directional guidance, leading participants to make choices aligned with the intended targets. This finding further validates the effectiveness of the individual manipulation experiment, demonstrating that small, customized datasets can achieve more efficient and precise human behavior manipulation.

## 6 DISCUSSION

Using recently developed counterfactual-based approach that generates synthetic visual stimuli along the perceptual boundaries of neural networks, our work provides new insight into variability of human percepts. First, we demonstrated that sampling along the perceptual boundaries of ANNs allows for the generation of stimuli that evoke diverse internal experiences among human observers. Second, through human experiments, we validated the effectiveness of our model in capturing the nuances of human perception. Third, by utilizing carefully designed controversial

stimuli, we selectively manipulated individual behavior, unveiling significant inter-individual differences in perceptual variability. Our study not only uncovers specific differences between humans and machines in the variability of their perceptual experiences but also offers effective tools for manipulating and predicting individual category judgments.

From the perspectives of cognitive science and neuroscience, our method significantly enhances the utility and flexibility of generated images in the study of human perception. Unlike the methods employed by Golan et al. (2020) and Feather et al. (2023), which reveal the disparities between model and human perception by generating images that strongly affect ANNs while having minimal impact on human cognition, our method is capable of influencing both models and human perception simultaneously. This dual impact allows for a nuanced counterfactual examination of the subtle differences in perceptual variability between the two. In contrast to the approaches taken by Veerabadran et al. (2023) and Gaziv et al. (2024), which focus on improving ANNs to produce images that can influence human perception, our use of diffusion models with prior distributions allows for broader applicability across various ANN models and perturbation methods. This also expands the range of image sampling, enabling sampling from high-noise areas like perceptual boundaries. Moreover, the incorporation of prior distributions ensures that our generated images more closely resemble natural images, enhancing their effectiveness in influencing human perception. With this significant improvement in the usability and flexibility of generated images, we successfully explored individual differences in human perception and opened the door for personalized manipulation, increasing the efficiency and scope of human perception studies.

From the perspective of computer science methodology, we have made significant improvements upon existing methods, opening new avenues for fields of AI for science and AI-human alignment. Drawing on the controversial stimuli from Golan et al. (2020) and adversarial perturbations from Veerabadran et al. (2023), we integrated these concepts with diffusion model priors to create two new loss-guiding methods: controversial guidance and uncertainty guidance. This enhancement increases the naturalness of the generated images and their influence on human perception. Additionally, inspired by works such as Jeanneret et al. (2023), Wei et al. (2024a), Wei et al. (2024b), Chen et al. (2023), Jeanneret et al. (2022), Vaeth et al. (2023), Atakan Bedel & Çukur (2023), we introduced counterfactual methodologies into the study of human perceptual variability, allowing us to explore this relatively under-researched area in greater depth. Our experiments demonstrate that the varMNIST dataset we generated significantly evokes human perceptual variability, providing a novel approach for aligning AI and human by harmonizing their perceptual variabilities. Furthermore, varMNIST can reveal individual differences among humans, enabling the generation of customized images that reflect these differences through ANNs aligned with individual participants.

Despite our progress in exploring human perceptual variability, several limitations remain. Our varMNIST dataset, generated by sampling along ANN perceptual boundaries, cannot fully capture human variability, especially influences like culture, as some ANNs are trained on data from specific groups. To address this, we plan to include participants from diverse cultural backgrounds for a more comprehensive understanding. Furthermore, the dataset's focus on handwritten digits, while effective for evoking perceptual variability, limits the exploration of broader visual phenomena. Expanding beyond object recognition to tasks like similarity judgments, emotion recognition, visual attention, and scene memory could offer deeper insights. However, exploring such complex tasks remains challenging given the limited number of trials available in individual behavioral experiments. Future work will incorporate natural images to better capture the complexity of human perceptual variability in more diverse and ecologically valid contexts.

In terms of aligning AI with humans, although ANNs finetuned with individual behavioral data showed a notable improvement in predicting perceptual variability, there remains a significant gap when compared to their performance in standard classification tasks. This indicates that perceptual variability is a promising but underexplored method for AI-human alignment, with ample room for improvement. To address this, we propose incorporating optimal experimental design Rainforth et al. (2024); Foster et al. (2019; 2021) into human experiments, using ANNs finetuned with individual behavioral data to generate customized images that maximize individual variability. These new behavioral data could then be fed back into the training of ANNs, dramatically improving AI-human alignment with fewer experimental trials. This approach would significantly increase the efficiency of human behavior data collection, reduce the cost of AI-human alignment, and accelerate the advancement of both cognitive science and artificial intelligence.

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

## A    DETAILS OF COLLECTING HUMAN PERCEPTUAL VARIABILITY

### A.1    CLASSIFIER GUIDANCE DIFFUSION MODEL

#### A.1.1    DIFFUSION MODELS.

Diffusion models Song et al. (2020); Karras et al. (2022) consist of two main phases: forward and reverse. The forward phase transforms an image into Gaussian noise over time $t \in [0, T]$, while the reverse phase reconstructs the image from noise by reversing this process. At any time $t$, the state $x_t$ is defined as:

$$x_t = a_t x_0 + b_t \epsilon_t, \tag{1}$$

where $a_t = \sqrt{\alpha_t}$, $b_t = \sqrt{1 - \alpha_t}$, $\alpha_t$ increases with $t$, and $\epsilon_t \sim \mathcal{N}(0, I)$. A neural network is trained to predict the added noise:

$$\min_\theta \mathbb{E}_{x_t, \epsilon_t} \left[ \|\epsilon_\theta(x_t, t) - \epsilon_t\|_2^2 \right], \tag{2}$$

where the loss depends on the noise and the probability distribution $p_t(x_t)$. The reverse process follows an ordinary differential equation (ODE):

$$\frac{dx_t}{dt} = f(t)x_t - \frac{g^2(t)}{2} \nabla_x \log p_t(x_t), \tag{3}$$

with $f(t) = -\frac{d \log a_t}{dt}$ and $g^2(t) = \frac{db_t^2}{dt} - 2\frac{d \log \sqrt{\alpha_t}}{dt} b_t^2$. This ODE enables the reconstruction of the image by reversing the noise-adding process.

The specific steps for both phases are determined by the sampling algorithm. We use the DDPM algorithm Ho et al. (2020), where the forward and reverse steps are represented as:

$$x_t = DDPM^+(x_{t-1}) \quad \text{and} \quad x_{t-1} = DDPM^-(x_t).$$

### A.1.2 CLASSIFIER GUIDANCE

Classifier guidance is also known as Training-free guidance. Using a diffusion model and the conditional information $y$, we define the conditional probability of the generative process as:

$$p(x_t|y) = \frac{p(y|x_t)p(x_t)}{p(y)}$$

where $x_t$ is the generated stimuli at time step $t$.

The gradient of this probability is calculated as follows:

$$\nabla_{x_t} \log p_t(x_t|y) = \nabla_{x_t} \log p_t(x_t) + \nabla_{x_t} \log p_t(y|x_t)$$

In the training-free approach, we utilize a network $f_\phi$ and define a loss function $\ell(f_\phi(x_t)), y)$ for conditional generation. Thus, we obtain:

$$\nabla_{x_t} \log p_t(y|x_t) = \nabla_{x_t} \ell(f_\phi(x_t), y)$$

In the reverse sampling process, we introduce a correction step:

$$x_{t-1} = DDPM^-(x_t) - \gamma \nabla_{x_t} \ell(f_\phi(x_t), y)$$

Therefore we can generate certain stimuli by designing the loss function $\ell$. To obtain stimuli that can disrupt human perception, we explored four potentially suitable approaches: uncertainty sampling and controversial sampling. High uncertainty sampling aims to generate stimuli that challenge the model's judgment, while controversial sampling seeks to produce stimuli that maximize the difference in probability distributions between two models.

## A.2 DETAILS OF GUIDANCE ALGORITHMS

### A.2.1 DETAILS OF TARGETED GUIDANCE

In targeted guidance, we specify directions for the guidance. For instance, at position (3, 5), uncertainty guidance directs towards both categories 3 and 5. For controversial guidance, classifier 1 is directed towards category 3, while classifier 2 is directed towards category 5. Similarly, at position (5, 3), uncertainty guidance directs towards both categories 5 and 3. For controversial guidance directs classifier 1 towards category 5 and classifier 2 towards category 3. To ensure balanced targeted guidance, we generate samples in multiples of 100 for each targeted guidance. This ensures that all stimuli corresponding to positions from 0 to 9 × 0 to 9 (i.e., covering all guidance directions) are included, thereby maximizing sampling uniformity. In the generation the guidance scale is set to 0.1, resampling steps is set to 5, and the inference steps is set to 50.

When generating stimuli using this guidance strategy, we ensure that each term in the loss function is effectively utilized. While this approach guarantees category-balanced sampling during the generation process, the final retained stimuli may not necessarily exhibit category balance. For the stimuli intended for human digit recognition experiments, we apply additional filtering to the generated images. Specifically, for uncertainty sampling, we require that the top two p-values exceed 0.4 and the digit surrogate score is above 0.5. For controversial sampling, we ensure that the classification outputs of both classifiers correspond to the intended guidance direction, with the highest p-value exceeding 0.9 and the digit surrogate score above 0.5. A detailed analysis of the filtered dataset derived from uniform sampling was performed, and the distribution of category counts is presented in Figure A.6. By comparing this distribution with the cognitive data shown in Figure A.12, a correlation can be observed.

### A.2.2 THE ROLE OF DIFFUSION PRIOR

In previous studies that employed generated images to investigate model and human perception ( Golan et al. (2020); Gaziv et al. (2024); Veerabadran et al. (2023); Feather et al. (2023)), a common issue was that the generated images lacked sufficient naturalness and failed to significantly influence human perception. This issue is particularly crucial within the context of our research objectives. Using previous methods often resulted in images that were unrecognizable to human participants,

leading to nearly random classification results (see Figure. A.2). Recent advances in adversarial examples and counterfactual explanations in machine learning (Jeanneret et al. (2023); Wei et al. (2024b); Chen et al. (2023); Jeanneret et al. (2022); Vaeth et al. (2023); Atakan Bedel & Çukur (2023)) have addressed this issue by employing diffusion models as regularizers to introduce prior information. This technique allows for the generation of natural images capable of influencing human perception.

Inspired by these advances, we utilize a classifier-free diffusion model as the core of image generation process. By sampling noise from the target dataset (MNIST) distribution and feeding it into the diffusion model for denoising, we effectively incorporate prior information. This approach enhances the naturalness of the generated images, making them more reflective of the real distribution of handwritten digits and thereby increasing their impact on human perception, as shown in Figure. A.2.

### A.2.3 Editing existing datasets vs. generating data from scratch

The process of adding MSE loss to the loss function can be seen as editing existing datasets. The MSE loss is used to constrain the pixel space of the stimuli. Without the MSE loss, the model is more likely to sample from distributions of stimuli that are very similar in pixel space within a certain class. We aim to enforce a constraint in the pixel space that encourages the stimuli to be closer to the original distribution of randomly sampled samples from the MNIST dataset. This approach is intended to enhance the diversity of the generated stimuli. We define $\alpha$ as the pixel-level restraint scale.

For uncertainty guidance with MSE constraint, we have:
$$\ell(f(x_t), y) = H(y|x_t) + \alpha||x_t - x_{ref}||^2,$$
where $H$ represents the entropy, $\alpha$ represents the strength of the MSE loss. For controversial guidance with MSE constraint, we have:
$$\ell(f(x_t), y) = D_{KL}(p(y_1|x_t), p(y_2|x_t)) + \alpha||x_t - x_{ref}||^2,$$
where $D_{KL}$ represents the KL divergence between two distributions. We conducted experiments on five models. For uncertainty sampling, we generated stimuli for each model, resulting in five groups of stimuli. In the controversial sampling experiments, we pitted the models against each other in pairs, creating ten groups of stimuli. However, this approach can be perceived as manipulating one class into another, which is similar to our goal of sampling along the decision boundaries of ANNs, but not exactly the same. In generation, the guidance scale is set to 0.1, resampling steps is set to 5, the inference steps is set to 50, and $\alpha$ is set to 50.

### A.2.4 Targeted guidance vs. untargeted guidance

Untargeted guidance focuses solely on increasing the variability of the generated images, without considering the overall distribution of the images. We adopted an untargeted guidance method to generate stimuli for the digit recognition experiment. To sample at the decision boundary of the model, we drew on previous research and proposed two guidance methods: uncertainty guidance and controversial guidance. Uncertainty guidance ensures that the generated images are as close as possible to the model's perceptual boundary by maximizing the entropy of the classification probability distribution of a single ANN model for the generated images, thereby obtaining images with high perceptual variability for the model. For uncertainty guidance, this can be represented as:
$$\ell(f(x_t), y) = H(y|x_t),$$
Where $H$ is the entropy, $y$ is the output probability of the neural network. Controversial guidance, on the other hand, utilizes two different ANN models and generates images that maximize the KL divergence between their classification probability distributions, thereby maximizing perceptual differences between the models. For controversial guidance, this can be represented as:
$$\ell(f(x_t), y) = D_{KL}(p(y_1|x_t), p(y_2|x_t)),$$
Where $D_{KL}$ is the KL divergence, $p(y_1|x_t)$ is the output probability of the first neural network, $p(y_2|x_t)$ is the output probability of the second neural network. In generation the guidance scale is set to 0.1, resampling steps is set to 5, and the inference steps is set to 50. Targeted and untargeted guidances are compared in Figure.A.4. Losses with and without MSE are also compared in Figure. A.3. The formula for the targeted guidance can be found at section 3.1 . The losses compared in this figure are untargeted losses.

### A.3 MODEL CONFIGURARATION AND TRAINING

#### A.3.1 DIFFUSION MODEL

**Configuration of DiT.** The Diffusion Transformer (DiT) (Peebles & Xie (2023)) is a Transformer-based diffusion model tailored for generative tasks. In our configuration, the model processes 28 × 28 grayscale images using a patch size of 2 × 2, resulting in patch embeddings transformed into sequences of hidden size 128, with 1 input channel and 10 output classes. The architecture includes 4 Transformer layers with 8 attention heads per layer and an MLP ratio of 4.0.

DiT incorporates Patch Embedding, Timestep Embedding, and Label Embedding modules. These embeddings are combined with fixed sinusoidal positional encodings to provide spatial and temporal context. AdaLN (Adaptive Layer Normalization) layers condition the model on timestep and label embeddings, with zero-initialized manipulation for training stability.

The model outputs spatial predictions through a final linear layer followed by an unpatching operation, restoring the input image dimensions. Classifier-free guidance is supported by computing conditional and unconditional outputs, enabling control over generated samples.

**Training of Diffusion Model.** For prior diffusion model, we use the MNIST dataset as the training dataset. The dataset consists of grayscale images of size 28 × 28, which are directly used without further resizing. The training process is conducted using a single GPU (NVIDIA GeForce RTX 4090) with the Adam optimizer.

The data is loaded into the training pipeline using a PyTorch DataLoader with a batch size of 128, andthe number of worker threads for data loading is set to 128. The model is trained for 150 epochs, with a learning rate of $1e - 4$ and an unconditional training rate of 0.1, and the weight decay is not applied. Dropout is applied to the class embedding with a probability of 0.1, while the model does not learn the variance (sigma).

#### A.3.2 CLASSIFIERS

Table 1: Configurations and MNIST Accuracy of Classifiers

| Model Name | Model Type | MNIST Accuracy (%) |
|---|---|---|
| ViT | Vision Transformer | 97.2 |
| VGG | Small VGG | 98.2 |
| CORNet | CORnet-Z | 98.9 |
| MLP | Multi-Layer Perceptron | 98.3 |
| LRM | Logistic Regression Model | 92.7 |

The classifier models were trained on the MNIST dataset using $28 \times 28$ grayscale images, normalized with the 'ToTensor' transformation. Training and testing sets were loaded with a batch size of 100, and the models were implemented with 5 different configurations (see Table. 1) to map input images to 10 output classes. Training was performed on an NVIDIA GPU using the AdamW optimizer ($lr = 1 \times 10^{-3}$) for 16 epochs, and CrossEntropyLoss function was used to compute the classification loss.

#### A.3.3 DIGIT JUDGMENT SURROGATE

**Training of digit judgment surrogate.** For the training of the digit judgment surrogate model, we constructed a dataset based on the results of the human digit judgment experiment. Specifically, for any given image, the frequency of participants responding "True" was taken as the probability of the image being judged as a digit. These images and their corresponding probabilities were then used to train the digit judgment surrogate. The dataset was split into a training set and a test set in a ratio of 8:2.

The surrogate model is based on the SmallVGG architecture, with a final output layer designed for regression tasks. The model was trained using the AdamW optimizer with a learning rate of 0.001, and the mean squared error (MSE) was used as the loss function. The training process lasted for 8

epochs with a batch size of 128. After each epoch, the validation loss was monitored, and the model with the best validation performance was saved for further evaluation.

**Performance of digit judgment surrogate.**   To ensure the validity of the digit judgment surrogate's predictions, we computed the correlation between the predicted scores and human scores. For any given image, the human score was defined as the frequency of participants responding "True," indicating the image is a digit, while the predicted score was the probability assigned by the model classifying the image as a digit.

As shown in Figure. A.8a, the Spearman rank correlation coefficient between the predicted scores and human scores is 0.8035. This indicates that the model's digit judgment is highly consistent with human. Additionally, Figure. A.8b presents image examples corresponding to different predicted scores. For scores of 0.10, 0.25, 0.50, 0.75, and 0.90, eight samples were randomly selected for each score. The examples reveal that as the predicted score increases, the images progressively resemble digits more closely. These results demonstrate that the digit judgment surrogate effectively simulates human digit judgment behavior.

### A.4    ONLINE HUMAN BEHAVIROAL MEASUREMENT

### A.4.1    DIGIT JUDGMENT

We use the initial synthetic dataset as experimental stimuli to measure human behavior in the judgment task (see Figure 2b (left)). The purpose of the experiment is to collect human judgments on whether any given image test is a digit, in order to filter out images that do not meet the standards of handwritten digits.

**Task paradigm**   Before the formal experiment, participants will first complete a pre-experiment. Each round of the pre-experiment consists of two stages. (1) Selection Stage: A test image appears at the center of the screen, with two buttons labeled "True" and "False" displayed below it. Participants are required to judge whether the image represents a number. (2) Feedback Stage: After making their choice, participants will receive feedback below the image indicating whether it is a number. The pre-experiment includes a total of 10 rounds, after which participants will proceed to the formal experiment. In formal experiment, participants performed multiple rounds of a choice task (see Figure. A.7). Each trial consisted of two phases: (1) Fixation Phase: A black cross was displayed at the center of the screen for 300 ms to direct participants' attention to the center. (2) Selection Phase: A test image appears at the center of the screen, with two buttons labeled "True" and "False" displayed below it. The positions of the buttons were fixed and remained unchanged throughout the trials. Participants were asked to judge whether the image represents a figure by selecting the corresponding button with the mouse or pressing the key on the keyboard (A represents True and D represents False). There was no time limit for responding. Each session of formal experiment comprised 500 trials, divided into two types: (1) Sentinel trials (n = 10), in which participants are shown a set of 10 pre-selected MNIST images, i.e., the correct response should be True. We screened participants based on their accuracy in the sentinel trials to ensure high-quality responses. (2) Random Trials (n=490), where images were randomly selected from the dataset, excluding the fixed images. The two trial types were presented in a random alternating order. No feedback was provided after participants made their selection. The experiment was programmed using JSPsych, with stimuli presented via the JSPsych-Psychophysics component.

**Human data collection**   The experiment got ethics approval from the local University. We recruited participants (N=400) and collected data through the NAODAO platform. Prior to the experiment, participants read an informed consent form detailing any potential risks associated with participation. Participants were allowed to withdraw from the experiment at any time. No personal identification information was collected. We only included data from participants with sentinel trial accuracy greater than 70%, resulting in data from 276 participants and 135240 trials involved in the following analyses.

### A.4.2 Digit recognition experiment

We used the filtered synthetic dataset as experimental stimuli to measure human behavior in a digit recognition task (see Figure. 2). The goal of the experiment was to collect the probability distribution of human choices for any given test image. In this task, participants were presented with ten possible choices, represented by the digits 0 to 9.

**Task paradigm** Participants performed multiple rounds of a category comparison task. Each trial consisted of two phases (see Figure. A.9): (1) Fixation Phase: A black cross was displayed at the center of the screen for 300 ms to direct participants' attention to the center. (2) Selection Phase: A test image appeared at the center of the screen, accompanied by ten labeled buttons below it, with labels ranging from 0 to 9. The positions of the buttons were fixed and remained unchanged throughout the trials. Participants were asked to identify the digit in the image by selecting the corresponding button with the mouse or pressing the number key on the keyboard. There was no time limit for responding.

Each session comprised 500 trials, where images were randomly selected from the dataset. No feedback was provided after participants made their selection. The experiment was programmed using JSPsych, with stimuli presented via the JSPsych-Psychophysics component.

**Human data collection** The experiment got ethics approval from the local University. The experiment collected behavioral data from 400 participants through the NAODAO platform, comprising 200,000 trials and 20,000 stimuli. Prior to the experiment, participants read an informed consent form detailing any potential risks associated with participation. Participants were allowed to withdraw from the experiment at any time. No personal identification information was collected. During data preprocessing, 154 participants were excluded based on Sentinel trials (accuracy < 0.7), leaving data from 246 participants (123,000 trials and 19,952 valid stimuli). Table 2 and Table 3 shows the stimuli distribution across guidance strategies and classifier. Using this cleaned dataset, we constructed a high perceptual variability dataset, varMNIST, which serves as a foundation for subsequent analysis and modeling.

## A.5 Additional dataset details

### A.5.1 Evaluation metrics

**Judgment distribution.** As shown in Figure. A.10 (top left), we evaluated the distribution of human judgments across the ten digit classes (0–9). The results indicate that the probabilities are relatively uniform, with all categories exhibiting values close to 0.1. Notably, digits 0, 6, and 9 were judged with slightly higher probabilities (around 0.15) compared to other digits, while digits 1 through 5 demonstrated lower probabilities (around 0.06).

**RT and entropy.** We further examined the relationship between response time (RT) and entropy to gain insights into the cognitive process underlying human judgments. RTs were predominantly distributed between 500 and 1500 ms, following a long-tail distribution, indicating that most decisions were made quickly, with a few requiring significantly more time (Figure. A.10 (top right)). The entropy of human judgments primarily concentrated near 0, reflecting high confidence in about half of the trials. Values between 0.5 and 2 also appeared, indicating uncertainty or ambiguity (Figure. A.10 (bottom left)). A positive correlation (Spearman rank correlation coefficient = 0.55) was observed between entropy and RT, suggesting that higher uncertainty in judgment often corresponds to longer decision times (Figure. A.10 (bottom right)).

**Classifier configurations influence the guidance outcome.** We evaluated how different classifier configurations affected the guidance outcome under controversial guidance conditions. The overall guidance success was determined by measuring the probability that participants selected digit $x$ when the model guided the judgment toward $x$. As shown in Figure. A.11 (left), the results show that CORNet and VGG achieved the highest success rates, both nearing 0.6, indicating their strong ability to influence human judgments. VIT and MLP followed with moderate success rates of approximately 0.3, while LRM had the lowest success rate at around 0.2, reflecting its weaker guidance capability.

Further analysis compared the guidance outcome differences between classifiers when used as adversarial pairs in controversial guidance (see Figure. A.11 (right)). CORNet and VGG consistently outperformed other classifiers, showing significantly higher success rates. In contrast, LRM exhibited the lowest success rates compared to other classifiers. These findings suggest that the choice of classifiers significantly impacts the effectiveness of controversial guidance, with certain architectures like CORNet and VGG being more effective at aligning human responses with their intended guidance.

**Guidance targets influence the guidance outcome.** We analyzed how different guidance targets influenced the guidance outcome, defined as the proportion of successful stimuli generated for each target pair. As shown in Figure. A.12, the results revealed significant variability across guidance targets. Target pairs such as (1, 7), (1, 2), and (4, 9) demonstrated the highest success rates, each exceeding 0.35. This suggests that these pairs may align better with human perceptual biases or model representations, leading to more effective guidance. Conversely, pairs such as (1, 8), (2, 9), and (7, 8) exhibited the lowest success rates, with values below 0.03, indicating greater difficulty in guiding these pairs. These findings highlight the importance of selecting appropriate guidance targets to maximize the effectiveness of the generated stimuli.

### A.5.2 ADDITIONAL VISUALIZATION RESULTS

We generated 900 images using both targeted and untargeted approaches under the guidance of uncertainty and controversial methods. A t-SNE analysis was conducted on the targeted and untargeted methods for both the controversial and uncertainty approaches. To ensure fairness, the t-SNE analysis was performed directly on the raw pixel space for dimensionality reduction. The results are shown in Figure A.5. It can be observed that the distribution is more uniform when targeted guidances are adopted.

## B  ADDITIONAL RESULTS OF PREDICTING HUMAN PERCEPTUAL VARIABILITY

### B.1  EFFECTS OF FINE-TUNING ACROSS CLASSIFIERS

**Prediction accuracy.** As shown in Figure. A.13, on MNIST, group/individual fine-tuning resulted in slight accuracy improvements for ViT and VGG, while CORNet and MLP showed no significant changes. LRM's accuracy decreased after fine-tuning, indicating limited generalization. On varMNIST, all classifiers exhibited significant accuracy gains after fine-tuning, highlighting the benefits of group and individual fine-tuning for datasets with high perceptual variability.

**Model and human entropy.** Figure. A.14 highlights the changes in correlation between model-predicted entropy and human behavioral entropy before and after fine-tuning. A positive correlation was observed across all baseline classifiers, indicating that even in the baseline condition, models capture human perceptual variability. Fine-tuning on varMNIST significantly enhanced this correlation, demonstrating improved alignment with human perceptual variability.

**Impact of Image Difficulty.** As shown in Figure. A.15, fine-tuned models outperformed baseline models across all entropy levels, confirming the general effectiveness of fine-tuning. For classifiers other than LRM, individual fine-tuned models achieved greater accuracy improvements on high-entropy images compared to group-tuned models, indicating that individual fine-tuning is particularly effective for challenging stimuli.

### B.2  MODEL FINE-TUNING

For group-level fine-tuning, The original classifier models were trained on the mixed (ratio = 1:1) MNIST , varMNIST datasets using $28 \times 28$ grayscale images, normalized with the 'ToTensor' transformation. For individual-level fine-tuning, the dataset is a mixture of varMNIST-i, varMNIST and MNIST at a ratio of 2:1:1 and the initial model is the group model. Training and testing sets were loaded with a batch size of 128, and the models were implemented with 5 different configurations (see Table. 1) to map input images to 10 output classes. Training was performed on an NVIDIA

GPU using the AdamW optimizer ($lr = 1 \times 10^{-3}$) for 16 epochs, and CrossEntropyLoss function was used to compute the classification loss.

### B.3 CLUSTERING ANALYSIS

There is a large variability in the subject's digit recognition behaviors, since participants differ in high-level factors such as culture, ethnicity, educational background, regional customs, and psychological states. We hypothesize that participants could be grouped into several clusters, with participants within the same cluster likely to exhibit similar perceptual variability. To test this hypothesis, we used each participant's subject-finetuned model to predict the behavior of all participants, and we calculated *inter-subject similarity matrix* based on the prediction results. The better the prediction performance, the higher the inter-subject similarity. As shown in Figure. A.16a, the similarity matrix between participants revealed the existence of eight distinct clusters. Furthermore, we observed that the subject-finetuned models performed better in predicting the behavior of participants within the same cluster (in-cluster) compared to those outside the cluster (out-cluster), as shown in Figure. A.16b. Our results indicate that the clustering is valid and that there are indeed high-level percept differences between participants.

## C DETAILS OF MANIPULATING HUMAN PERCEPTUAL VARIABILITY

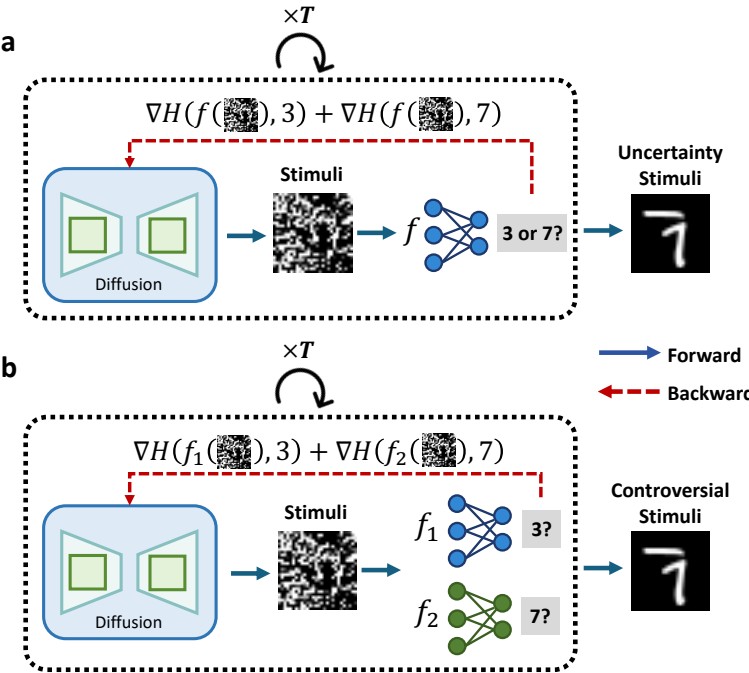

Figure A.1: **Guidance methods.** (a) The uncertainty guidance. It utilizes the classification uncertainty of the generated images from model $f$ to guide the diffusion model in generating stimuli toward specific directions. Model $f$ ensures the image is a digit. (b) The controversial guidance. It employs the classification differences between the generated images from model $f_1$ and model $f_2$ to guide the diffusion model in generating stimuli toward specific directions.

Table 2: Stimuli Counts before Experiment

| Guidance Strategy | Classifier | Stimuli Count |
|---|---|---|
| Controversial | CORNet_LRM | 1000 |
| | CORNet_MLP | 1000 |
| | MLP_LRM | 1000 |
| | VGG_CORNet | 1000 |
| | VGG_LRM | 1000 |
| | VGG_MLP | 1000 |
| | ViT_CORNet | 1000 |
| | ViT_LRM | 1000 |
| | ViT_MLP | 1000 |
| | ViT_VGG | 1000 |
| Uncertainty | CORNet | 2000 |
| | LRM | 2000 |
| | MLP | 2000 |
| | VGG | 2000 |
| | ViT | 2000 |
| Sum | | 20000 |

Table 3: Stimuli and Trial Counts after Experiment

| Guidance Strategy | Classifier | Stimuli Count | Trial Count |
|---|---|---|---|
| Controversial | CORNet_LRM | 997 | 5766 |
| | CORNet_MLP | 996 | 5688 |
| | MLP_LRM | 997 | 5684 |
| | VGG_CORNet | 995 | 5806 |
| | VGG_LRM | 994 | 5767 |
| | VGG_MLP | 999 | 5823 |
| | ViT_CORNet | 997 | 5865 |
| | ViT_LRM | 995 | 5949 |
| | ViT_MLP | 999 | 5811 |
| | ViT_VGG | 999 | 5881 |
| Uncertainty | CORNet | 1994 | 11631 |
| | LRM | 1992 | 11668 |
| | MLP | 1997 | 11849 |
| | VGG | 1996 | 11710 |
| | ViT | 1996 | 11817 |
| Sum | | 19943 | 116715 |

**Guidance without prior** **Guidance with prior** **VAE**

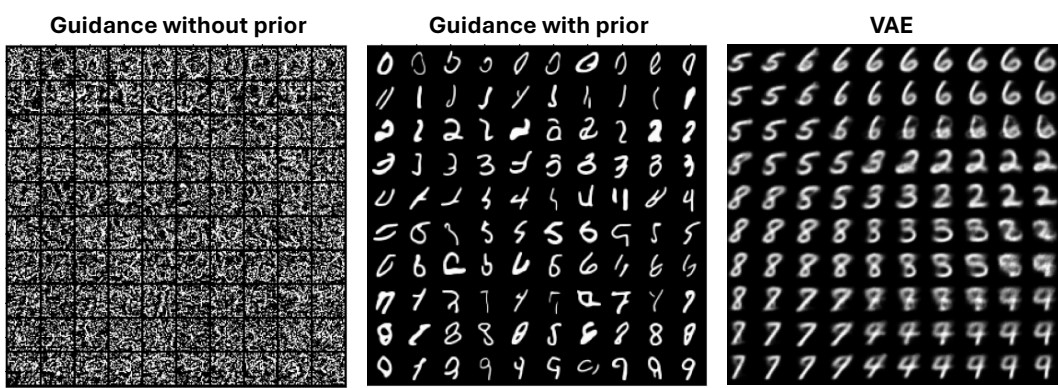

Figure A.2: **Comparison of our method with other approaches.** Images generated by a diffusion model without prior distribution exhibit severe noise. Images produced by a Variational Autoencoder (VAE) show minimal differences and are generally blurry. Our method (with prior), however, yields images that are not only clear and noise-free but also exhibit substantial diversity.

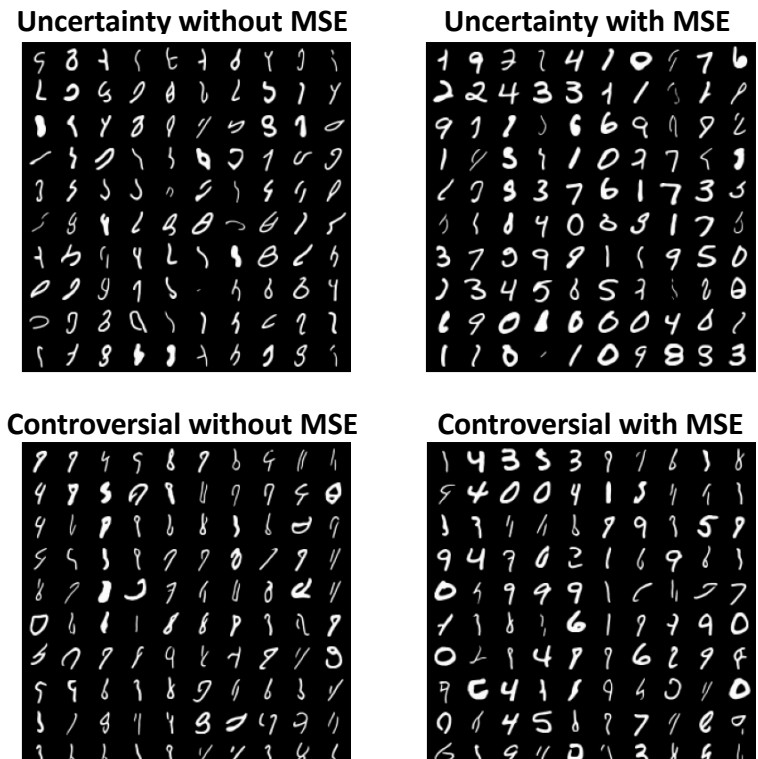

Figure A.3: **Comparison of generated images with/without MSE loss.** The losses here are all untargeted.

**Untargeted uncertainty**   **Targeted uncertainty**

**Untargeted controversial**   **Targeted controversial**

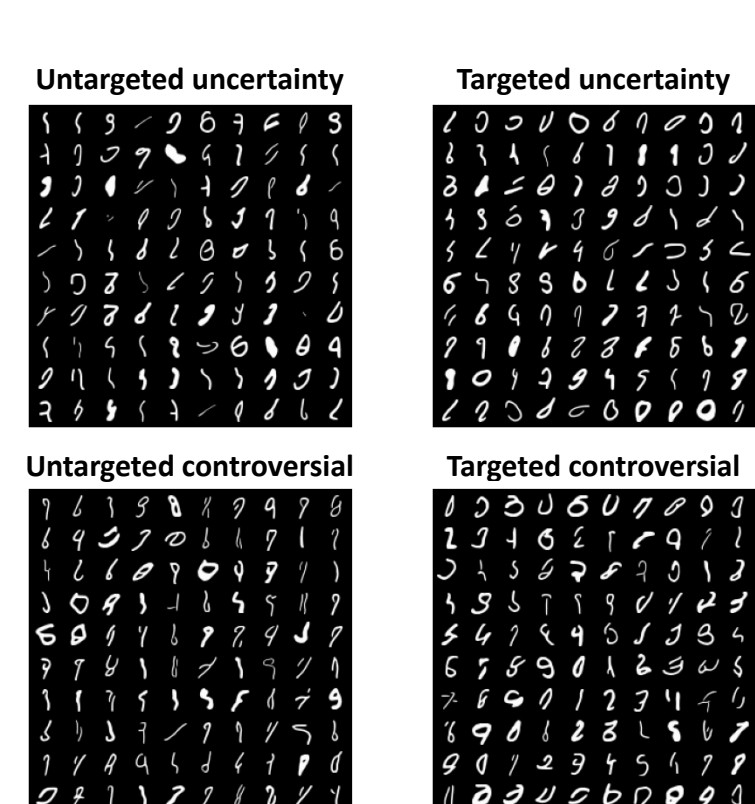

Figure A.4: **Examples of generated stimuli.**

**Controversial**   **Uncertainty**

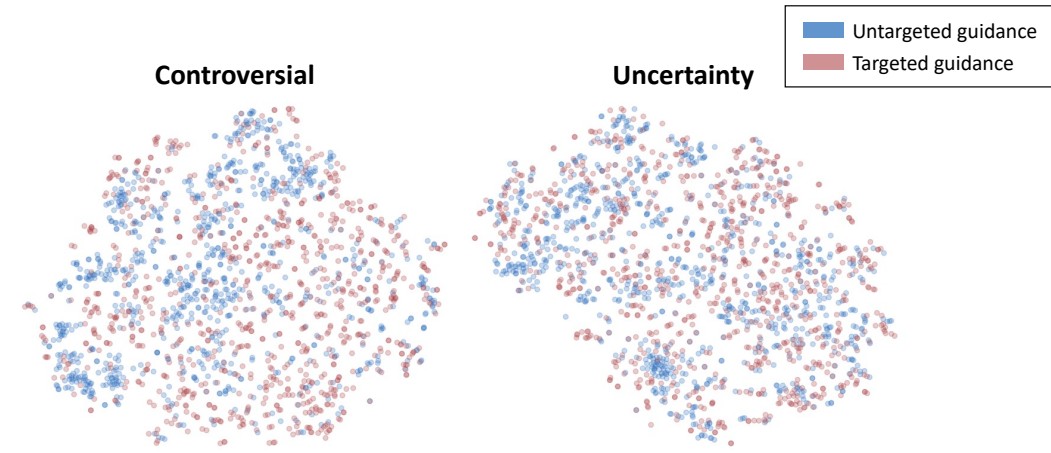

Figure A.5: **t-SNE analysis on the pixel space for the generated images of different guidance methods.** From the figure it is obvious that stimuli from targeted sampling are distributed more uniformly.

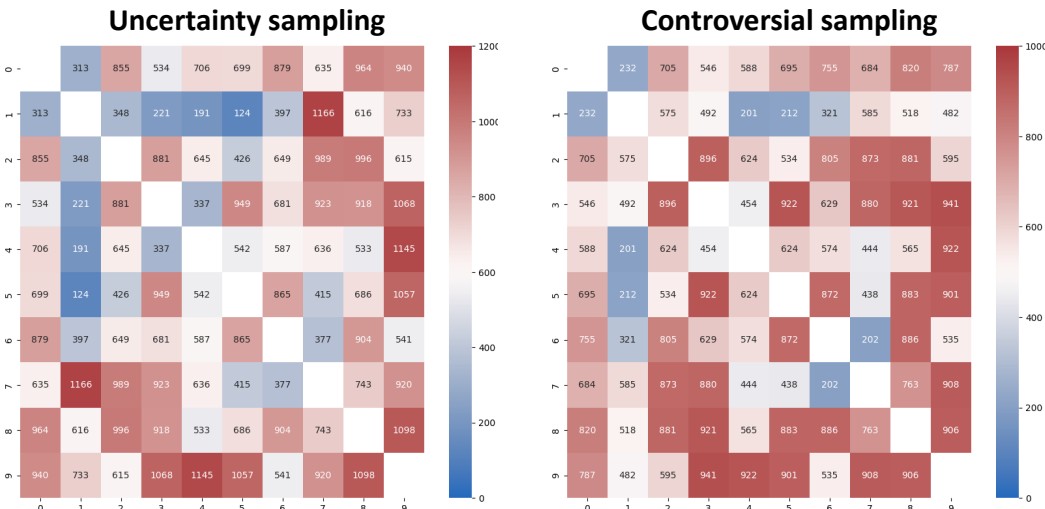

Figure A.6: **Category distribution in uncertainty sampling and controversial sampling.** Certain target pairs yield a higher number of stimuli that successfully pass the filtering criteria. For instance, target pairs such as (4, 9) and (8, 9) consistently produce more valid stimuli under both uncertainty and controversial sampling. In contrast, pairs like (1, 5) and (1, 3) result in significantly fewer stimuli meeting the filtering requirements.

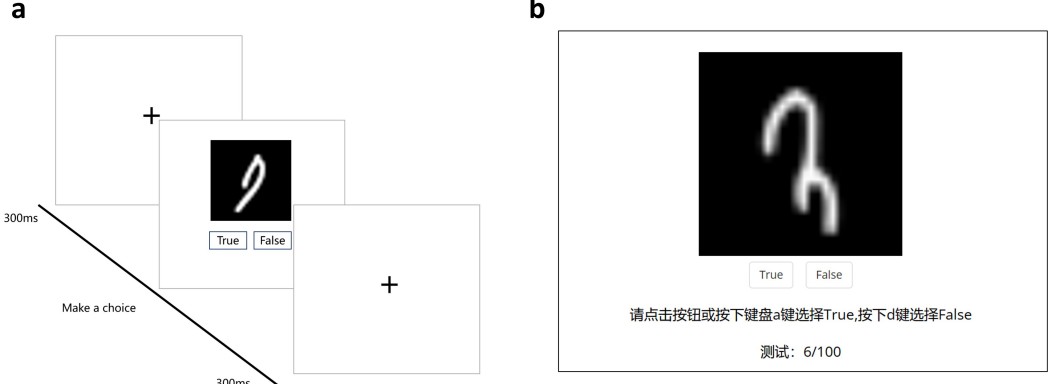

Figure A.7: **Human digit judgment experiment procedure.** In each trial, participants first observe a fixation cross ("+") for 300 milliseconds. Following the fixation, a stimulus image is presented along with 2 clickable buttons labeled "True" and "False". Participants are instructed to judge whether the image represents a digit and either click the corresponding button or pressing the key on the keyboard (A represents True and D represents False). The images shown to participants are generated by our model. After each selection, no feedback is provided, and the next trial begins immediately. Each participant first performed 10 rounds of pre-experiments with feedback, followed by 500 formal trials without feedback, including 10 sentinel trials and 490 random trials.

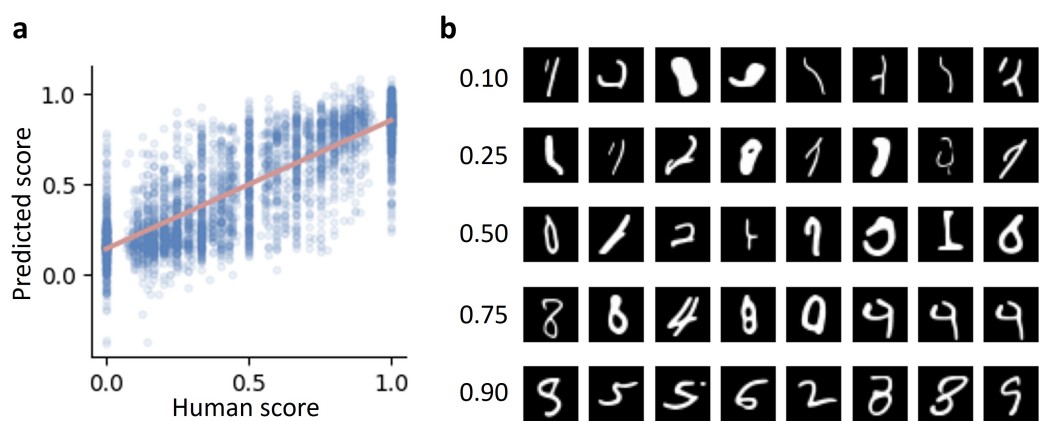

Figure A.8: **Performance of digit judgment surrogate.** (a) The predicted scores and human scores show a strong correlation. For any given image, the human score is defined as the frequency of participants answering "True" for the image being a digit, while the predicted score is the model's probability of classifying the image as a digit. The Spearman rank correlation coefficient between the two scores is 0.8035. (b) Examples of images with different scores. For predicted scores of 0.10, 0.25, 0.50, 0.75, and 0.90, 8 samples are randomly displayed for each score. As the score increases, the images increasingly resemble digits.

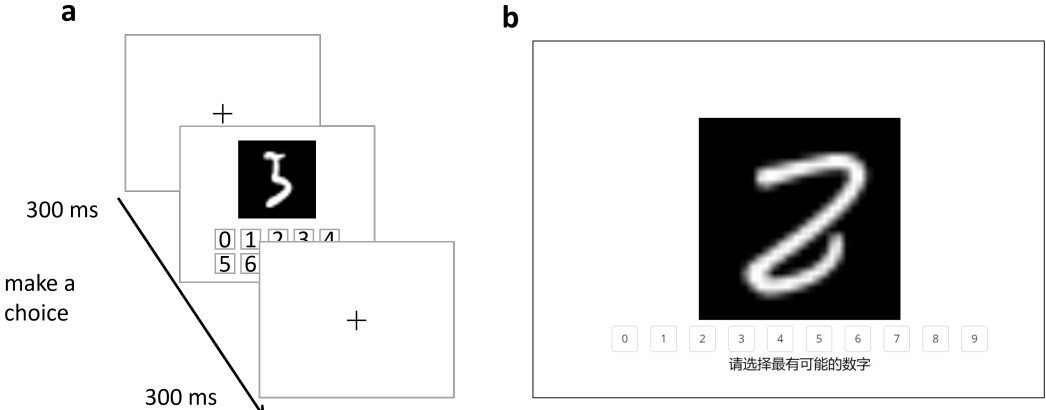

Figure A.9: **Human digit recognition experiment procedure.** In each trial, participants first observe a fixation cross ("+") for 300 milliseconds. Following the fixation, a stimulus image is presented along with 10 clickable buttons representing the digits 0 to 9. Participants are instructed to identify the most likely digit represented by the image and either click the corresponding button or press the corresponding number on the keyboard. The images shown to participants are generated by our model. After each selection, no feedback is provided, and the next trial begins immediately. Each participant completes a total of 500 trials, consisting of 10 sentinel trials and 490 random trials.

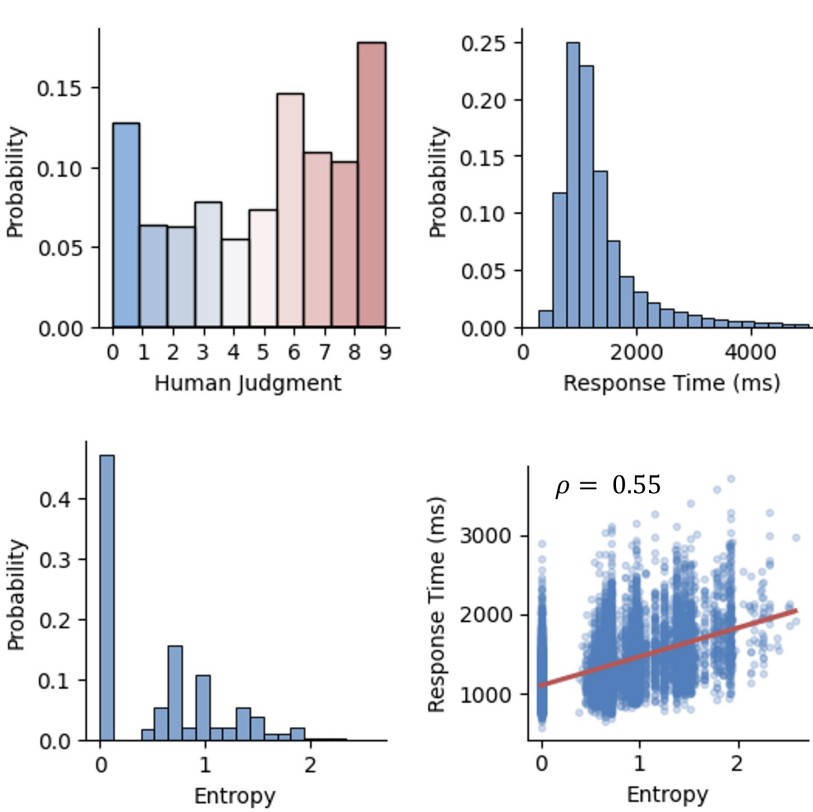

Figure A.10: **Behavioral results of the Digit recognition task.** Top right: In the digit recognition task on the varMNIST dataset, human judgment probabilities are relatively uniform, with values close to 0.1 for each category. Among these, digits 0, 6, and 9 have relatively higher probabilities, while digits 1 to 5 have lower probabilities. Top left: Human response times for the digit recognition task are concentrated between 500 and 1500 ms, showing a long-tail distribution. Bottom right: The entropy of human judgment results is primarily distributed around 0, with additional values observed between 0.5 and 2. Bottom left: Entropy and response time exhibit a positive correlation, with a Spearman rank correlation coefficient of 0.55.

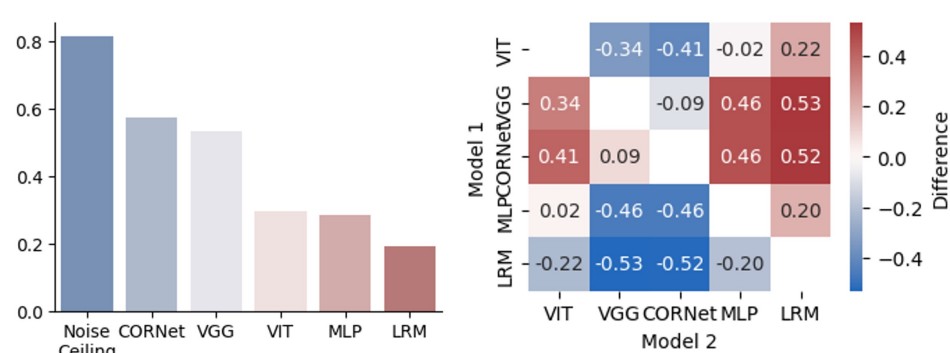

Figure A.11: **Guidance outcome under different classifiers configurations.** Left: Overall controversial guidance outcomes for different classifiers. Under controversial guidance conditions, the success rate is measured as the probability that participants chose digit $x$ when the model guided the judgment to $x$. CORNet and VGG achieved the highest success rates, nearing 0.6, followed by VIT and MLP with success rates of approximately 0.3. LRM had the lowest success rate at around 0.2. Right: Differences in guidance outcomes among classifiers during controversial guidance (when using two classifiers as adversarial classifiers, the difference in the guidance outcome of one classifier and the other). CORNet and VGG exhibited significantly higher success rates compared to other classifiers, while LRM showed notably lower success rates than the rest.

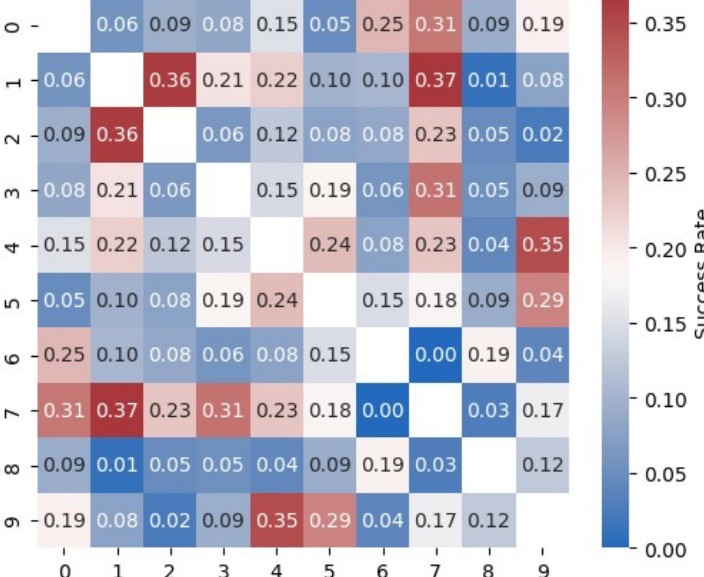

Figure A.12: **The success rate (proportion of successful stimuli) varies significantly across different guidance targets.** Pairs such as (1, 7), (1, 2), and (4, 9) achieve the highest success rates, exceeding 0.35. In contrast, pairs such as (1, 8), (2, 9), and (7, 8) have the lowest success rates, falling below 0.03.

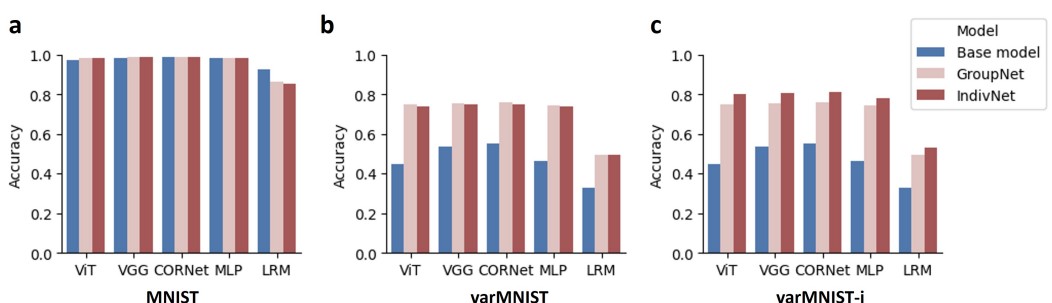

Figure A.13: **Correlation between model entropy and human behavior.** (a) Prediction accuracy on MNIST before and after fine-tuning for different classifiers. ViT and VGG show slight improvements in accuracy after group/individual fine-tuning, CORNet and MLP exhibit no significant changes, while LRM experiences a decrease in accuracy post-fine-tuning. (b) Prediction accuracy on varMNIST before and after fine-tuning. All five classifiers demonstrate substantial improvements in accuracy after group/individual fine-tuning. (c) Prediction accuracy on varMNIST-i before and after fine-tuning. All five classifiers show moderate improvements in accuracy after group/individual fine-tuning.

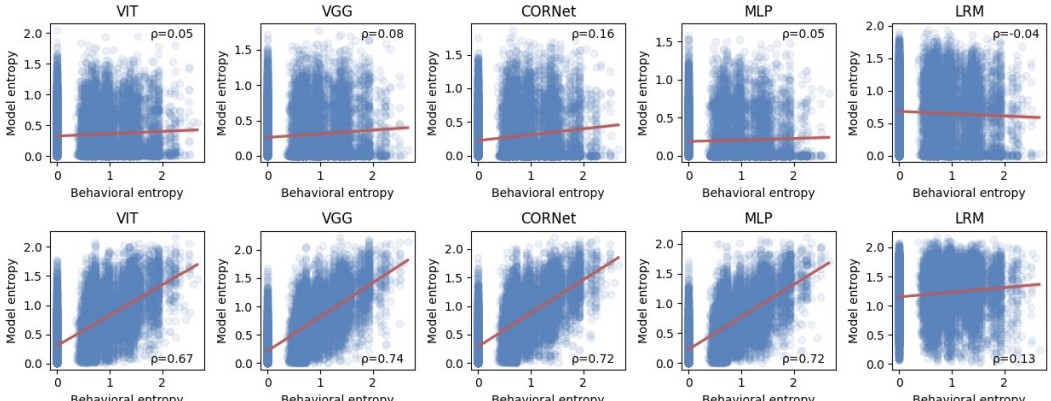

Figure A.14: **Correlation between model entropy and human behavior.** (a) Positive correlation between the entropy calculated from participants' behavior and the entropy predicted by the model for visual stimuli across five models. Each blue dot represents an image stimulus, and the red line shows the fitted result. (b) Significant improvement in the correlation between behavioral entropy and model-predicted entropy after fine-tuning on varMNIST, across five models.

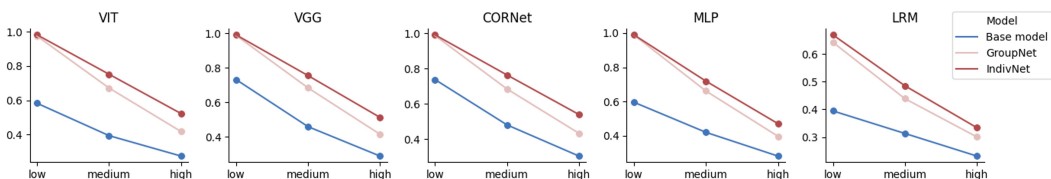

Figure A.15: **Correlation between model entropy of different classifiers and human behavior.** Prediction accuracy of different classifiers on images with varying entropy levels before and after fine-tuning. For all five classifiers, fine-tuned models show significant improvements in accuracy across all entropy levels compared to the baseline models. For the four classifiers other than LRM, the improvements of individual fine-tuned models over group-fine-tuned models are primarily observed on high-entropy images.

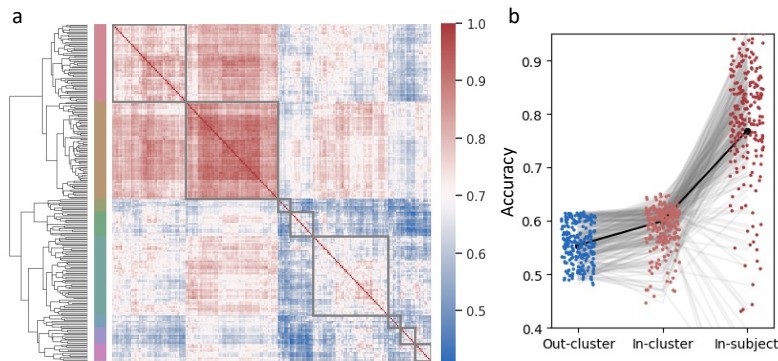

Figure A.16: **Subject clustering analysis.** (a) Subject similarity matrix and clustering results. The subject-finetuned model was used to predict the entire varMNIST dataset, and similarity between subjects was computed based on their prediction results. The left axis and gray boxes indicate subjects belonging to the same cluster, with a total of eight clusters. (b) Performance of the subject-finetuned model in predicting data from different groups: out-cluster, in-cluster, and in-subject correspond to different clusters, the same cluster, and the subject itself, respectively. Each point represents the average prediction performance of a subject on data from the corresponding group, and The black line represents the average of all subjects.

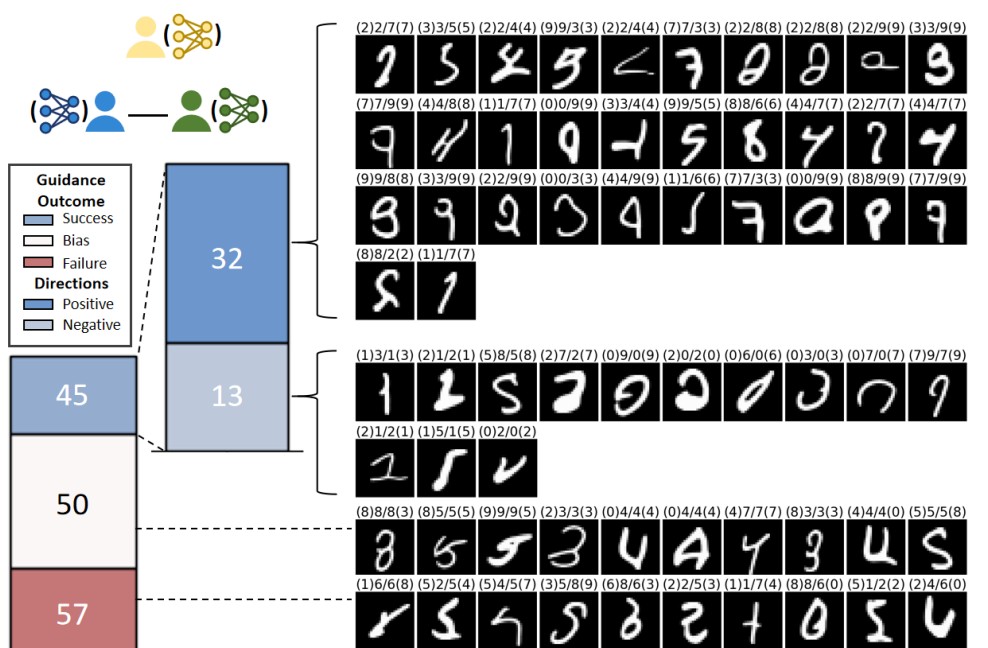

Figure A.17: **Examples of manipulation stimuli for subject 1 and subject 2.** The left part of the figure shows the actual numbers of each category of stimuli. The real stimuli used to manipulate the subjects are shown on the right. The choices of the subjects are in the middle, with the guidance label marked in parentheses. All positive and negative examples are presented, along with 10 typical bias and failure cases.

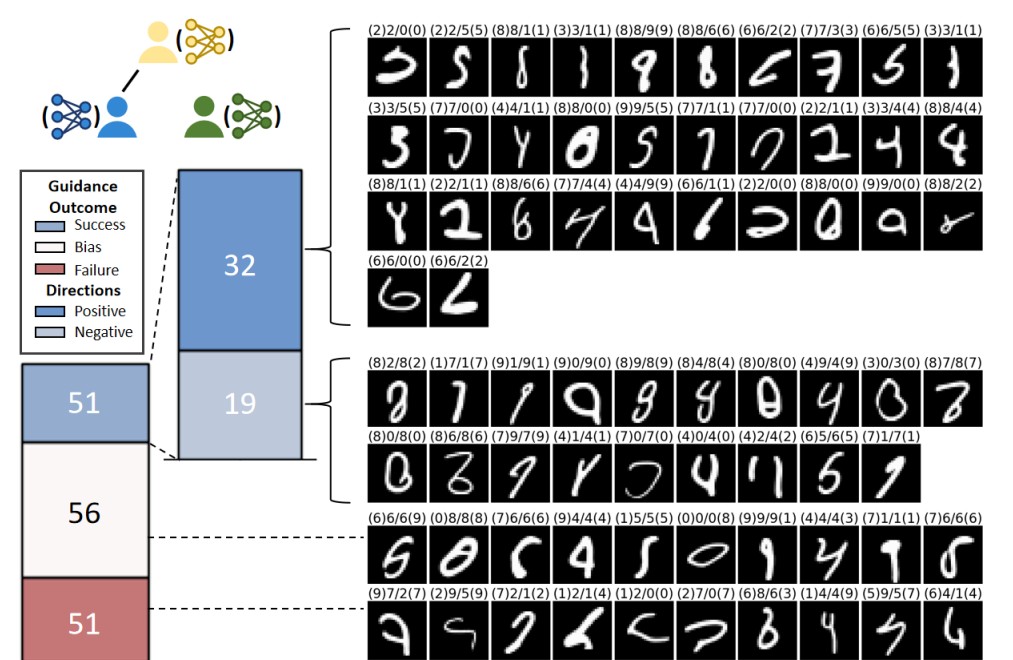

Figure A.18: **Examples of manipulation stimuli for subject 1 and subject 3.** The structure is the same as Figure A.17.

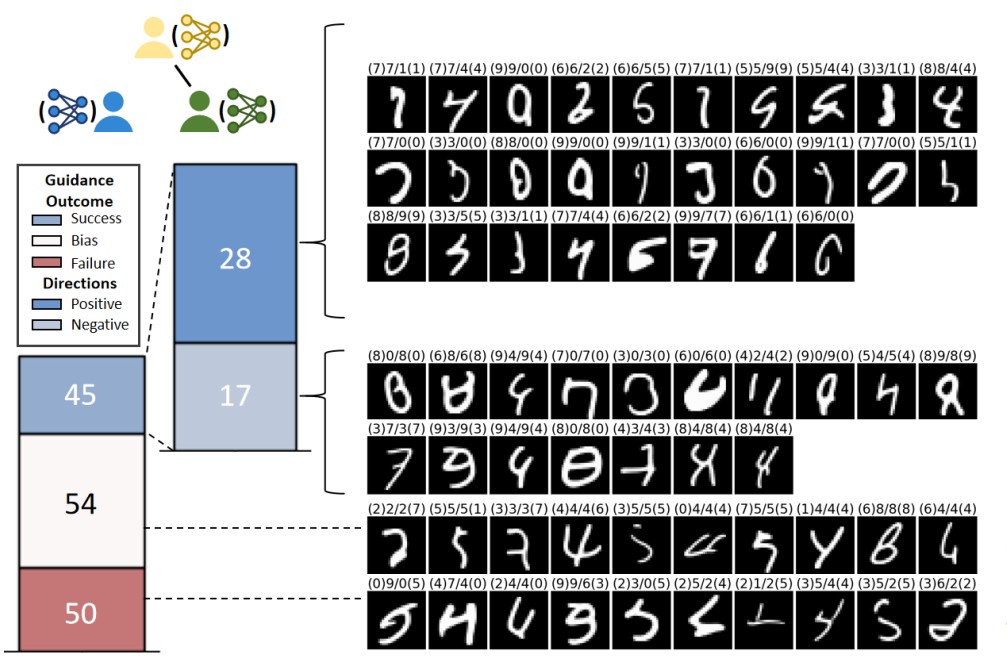

Figure A.19: **Examples of manipulation stimuli for subject 2 and subject 3.** The structure is the same as Figure A.17.

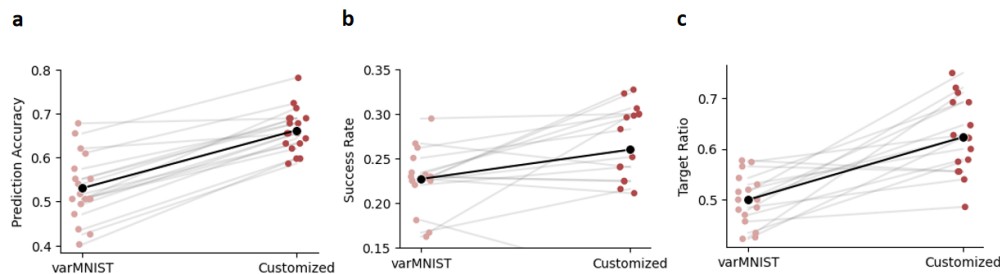

Figure A.20: **Detailed analysis of each subject pair.** (a) Prediction accuracy of individual models trained on in-lab participants, showing consistent improvement after fine-tuning. The original models are group models fine-tuned on varMNIST. (b) Comparison of guidance success rates between varMNIST and customized stimuli, indicating notable improvement for the majority of subject pairs. (c) Target ratio comparisons on varMNIST and customized stimuli, demonstrating an increase across nearly all subject pairs.

