# OpenReview forum: "Synthesizing Images on Perceptual Boundaries of ANNs for Uncovering and Modulating Individual Human Percepts"
_ICLR.cc/2025/Conference — Submitted to ICLR 2025_

### Official Review · Reviewer_vi4H · 2024-10-22

**Soundness:** 3
**Presentation:** 3
**Contribution:** 2
**Rating:** 6
**Confidence:** 3

**Summary:**

This paper presents a counterfactual generation-based method to study subject-level perceptual variability in humans. To automatically and scalably synthesize stimulus samples for human studies, the authors propose a generative model that samples images from the perceptual boundaries of artificial neural networks (ANNs) and integrates a post-hoc data curation procedure involving human input. The generative model guides the diffusion model using two approaches: (i) uncertainty guidance, which maximizes the entropy of the ANN’s predictions, and (ii) controversial guidance, which maximizes the prediction discrepancy between two different ANN models. Using this method, the paper introduces a synthetic dataset, varMNIST, to systematically investigate subject-level perceptual variability in three areas: (i) human behavioral analysis in image recognition tasks that evoke perceptual variability, (ii) alignment between ANN and human perception, and (iii) subject-level perceptual variability in response to controversial stimuli.

**Strengths:**

1. The paper provides interesting results from human studies, particularly in aligning perceptual boundaries between ANNs and human perception.
2. It introduces a novel end-to-end system to generate datasets for cognitive studies on human perception and its relationship with vision ANNs.
3. The paper conducts diverse empirical evaluations, complemented by human studies, to analyze the proposed datasets and uncover insights into human perception.

**Weaknesses:**

1. The technical contribution of uncertainty and controversial guidance is marginal, as these concepts already exist in prior works, and the paper does not offer significant system engineering innovations.
2. The post-hoc analysis of human studies lacks rigor. The findings on human perception are not supported by statistical significance, and the empirical results are not discussed in depth. For example, the paper notes inter-subject and subject-group differences in controversial stimuli but does not explore the factors contributing to these perceptual differences.

**Questions:**

- The formulation in lines 185 and 192 seems inconsistent. The left-hand side introduces $f_{\phi}(\cdot)$, but the right-hand side does not include $f_{\phi}$​ or the parameter $\phi$. It would be helpful to rewrite these equations more formally.
- The cluster analysis results in eight distinct clusters, suggesting high perceptual variability among participants. However, what characteristics of the participants contribute to this variability? Is it due to demographic attributes, experimental environment, or purely biological differences? Additionally, why does the analysis yield eight clusters?
- When interpreting results from human experiments, statistical significance should be considered. For example, in Figure 5, the difference between the "finetuned" and "subject-finetuned" results is minimal, yet the paper describes this as a "significant difference in perceptual variability across participants." To ensure validity in human studies, a stricter approach with p-values would be appropriate.
Could you clarify when uncertainty guidance and controversial guidance are used? Are they applied simultaneously or separately to generate images? In line 255, the paper states that images are generated using both guidance methods. Does this mean they are integrated to create a single image, or are they used separately to generate different subsets of images? If the latter, has there been any quantitative ablation study comparing their effectiveness in generating ambiguous samples for human perception?
- As noted in the paper, many prior works on adversarial examples and counterfactual explanations also leverage diffusion models to create natural samples. This setup is not novel.
- The technical novelty of this work is limited. The use of classifier guidance in diffusion models for conditional generation is a well-known technique, and post-processing samples with human evaluation is a standard dataset curation method. While the paper highlights uncertainty guidance, controversial guidance, and the end-to-end system as contributions, the controversial guidance was previously introduced by Golan (2020), and uncertainty guidance resembles untargeted adversarial attacks, which also occur at the classifier’s decision boundary, yielding high-entropy predictions. From a systems perspective, the paper does not offer system-level engineering innovations to enhance the generation process. Therefore, the main contribution lies in the problem addressed and its application to cognitive studies of human perception.
- There is a typo in line 310: "perveptual" should be corrected to "perceptual."

---

> ### Author Response · Authors · 2024-11-25
> **Response to Reviewer vi4H**
>
> We sincerely thank you for your thoughtful and detailed feedback on our work. Your comments have been instrumental in improving the quality of our manuscript. We have thoroughly addressed the points you raised and incorporated major revisions as detailed in the **Global rebuttal** section.
>
> `W1. Technical contribution`
>
> Thank you for pointing out this issue. We have revised the manuscript to better highlight our innovations (see **Global rebuttal 3**). As shown in our updated **Introduction** and **Figure 1**, the primary innovation of our work lies in constructing a systematic research paradigm. This paradigm leverages generated images to reveal human perceptual variability, achieves human-AI alignment at the individual level, and enables targeted modulation of specific participants’ behavior.
>
> We proposed a systematic process from **collection** to **prediction** to **manipulation**, conducting groundbreaking and in-depth research into human perceptual variability. While the techniques of uncertainty and controversial guidance have been explored in prior studies, we are the first to apply these methods to the scientific problem of "human perceptual variability."
>
> Additionally, we introduced new behavioral experiments (see **Global rebuttal 1**) to further validate our generative model. These experiments demonstrate that the stimuli generated by our model can elicit significant controversial behavior between participants (e.g., **ensuring Participant A selects digit 4 while Participant B selects digit 5 on the same stimulus**). This is the **first time** such precise cross-participant modulation has been achieved in visual perception tasks. We believe this work will provide significant insights for future research in human behavior and studies on AI-human alignment.
>
> `W2. The post-hoc analysis`
>
> Thank you for your valuable suggestions. We have made major revisions to the manuscript to address the issues you raised. Please refer to the **Global rebuttal 2** section for details. Our analysis of human experiment on varMNIST now focuses on **Fig. 3 and 4** in the updated manuscript, where we provide an in-depth discussion of experiment results and human-AI alignment.

---

> ### Author Response · Authors · 2024-11-25
> **Response to Reviewer vi4H (pt. 2)**
>
> `Q1. "The formulation in lines 185 and 192..."`
>
> Thanks for your correction, we have rewritten the formulas more formally (see **Sec. 3.1**).
>
> `Q2. High perceptual variability among participants`
>
> Both clustering analysis and behavioral analysis (e.g., **Fig. 3 and A.8**) indicate significant perceptual variability among participants on varMNIST. Factors such as demographic characteristics, experimental environments, or purely biological differences could potentially contribute to this variability, but these factors alone are unlikely to fully explain the complexity of perceptual behavior. For instance, two participants may consistently disagree on various stimuli across different digits, which cannot be attributed to a single factor. This is why we introduced individual models to align with specific participant behaviors. These models enable the generation of controversial stimuli between any two participants, providing an opportunity to explore differences in cognitive boundaries between individuals (see **Fig. 5**).
>
> As for the choice of the number of clusters, it was primarily determined for visualization purposes without specific theoretical justification. Regardless of the exact number, the results consistently demonstrate meaningful intra-cluster and inter-cluster differences. Additionally, we found that analyzing the manipulation experiments offered a clearer explanation of perceptual variability among participants. Therefore, in the updated manuscript, we focused on a detailed analysis of the manipulation experiments (see **Sec. 5**) and moved the clustering analysis to the appendix. For further details, please refer to **Global rebuttal 3**.
>
> `Q3. Statistical significance`
>
> Thank you for raising this concern. In response, we re-ran the human experiments (**Global rebuttal 4**) and model fine-tuning to ensure statistical rigor. The updated results demonstrate that the individual fine-tuned model achieved an average accuracy improvement of nearly 5% compared to the group fine-tuned model (see **Fig. 4b**). Across all 246 participants, accuracy improved for 241 participants and decreased for only 5, yielding a p-value approaching 0 for this improvement.
> It is also worth noting that each participant completed experiments with only 500 samples, a small dataset compared to MNIST (70,000 samples) and varMNIST (20,000 samples). Despite this limitation, our findings demonstrate that even with a small sample size, we successfully captured perceptual variability among individuals.
>
> In addition, we computed the statistical significance for our new manipulation experiments (**Fig. 5c**). With 18 participants, our manipulation results demonstrated statistical significance, further validating the effectiveness of our findings.
>
>
> `Q4. clarify uncertainty guidance and controversial guidance`
>
> Uncertainty guidance and controversial guidance are used separately to generate different subsets of images (see **Table 2 and 3**). We also conducted a quantitative analysis to compare their effectiveness in generating ambiguous samples for human perception (see **Fig.3b, A.11, A.13, A.14**).
>
> `Q5. Technical novelty`
>
> Yes, our work is applicable to cognitive studies and human-AI alignment. However, as noted in **W1**, the novelty of our work does not primarily lie in combining existing algorithms. Instead, it lies in proposing a **systematic process** from **collection** to **prediction** to **manipulation** to study **human perceptual variability**. Compared to exploring decision boundaries between AI models, investigating the differences in decision boundaries among humans is a far more challenging and significant problem.
>
> The human brain is a black box, and we can only collect extremely limited data from individuals, making this task particularly complex. To address this challenge, we proposed methods tailored to generate stimuli with high perceptual variability for human participants (**Sec. 3.1, 3.2**), conducted extensive behavioral experiments (**Sec. 3.3**), and ultimately demonstrated the ability to manipulate human behavior effectively. Our results validate that it is possible to reveal **human perceptual variability** using limited individual data. Although this was achieved within the relatively simple context of digit recognition tasks, it represents a groundbreaking step in achieving precise behavioral manipulatio—something previously demonstrated only in AI systems.

---

> > ### Comment · Reviewer_vi4H · 2024-12-01
> >
> > I thank the authors for their thorough response to my comments. My primary concern regarding the lack of post-hoc analysis and statistical significance measures has been addressed with additional experiments and more in-depth discussions, which I  appreciate. As a result, I raised my score.
> >
> > While I acknowledge the paper's main contribution in establishing a systematic framework for conducting a specific type of human studies, I remain somewhat unconvinced about the level of technical novelty. I encourage the authors to further articulate how their work pioneers an important domain of human perception studies in this regard.

---

> > > ### Author Response · Authors · 2024-12-02
> > >
> > > We are pleased to see that you recognize the improvements in our work, and we sincerely appreciate your acknowledgment of our contribution to "establishing a systematic framework for conducting a specific type of human studies." Below, we provide our detailed responses to better clarify our contributions and address your comments.
> > >
> > > **1. Importance and Challenges of the Research Problem**
> > >
> > > As mentioned in the **Introduction**, our core research problem is to uncover the differences in internal experiences among individuals when exposed to the same external stimuli (i.e., human perceptual variability). This problem has been a long-standing question in neuroscience and cognitive science. Furthermore, it is equally critical in the fields of AI-human alignment and human-computer interaction, as building personalized AI models and enabling precise interactions rely heavily on understanding and leveraging these perceptual differences.
> > >
> > > However, due to the significant data requirements of AI models, which often necessitate large-scale data collection across diverse participants, individual variability and its implications for personalized models have often been overlooked. In recent years, the proliferation of AI tools and generative models in cognitive science has inspired new methodologies. Nonetheless, as highlighted in our **Related Works**, research specifically addressing human perceptual variability remains sparse, underscoring the necessity and innovation of our work.
> > >
> > > **2. Contributions to the Field**
> > >
> > > As previously mentioned, we proposed a systematic process from **collection** to **prediction** to **manipulation**, conducting groundbreaking and in-depth research into human perceptual variability. Below, we provide a more detailed explanation of these contributions:
> > >
> > >  - 2.1 **Eliciting Human Perceptual Variability Through Generated Samples.** We demonstrated through extensive behavioral experiments that sampling along the perceptual boundaries of ANNs can produce stimuli that elicit significant human perceptual variability. As you noted, prior studies have employed diffusion models to create adversarial examples and counterfactual explanations. However, these studies primarily focused on comparisons between models or between models and humans, without generating or experimentally validating samples that differentiate between individual human participants (see **Section 3**).
> > >
> > >  - 2.2 **Achieving Individual Alignment by Predicting Human Perceptual Variability.** We showed that individual perceptual differences in visual decision-making tasks are significant and can be revealed through limited experiments. By fine-tuning models with high-variability images, we achieved alignment between AI models and individual human perceptual experiences. High-variability images highlight subtle individual differences often obscured in traditional datasets, improving data collection efficiency and enabling individual alignment with fewer samples (see **Section 4**).
> > >
> > >  - 2.3 **Revealing and Manipulating Perceptual Variability Using Individual Models.** Building on predictions of human perceptual variability, we pioneered personalized manipulation experiments. Using controversial stimuli designed with individual models, we achieved—for the first time—controlled experiments where the same stimulus elicited different, expected responses from two participants. This highlights the potential of personalized AI models for precise modulation of human perception (see **Section 5**).
> > >
> > > As mentioned in **Global Rebuttal 3**, we have already made substantial updates to the manuscript to clarify these contributions. Additionally, we will revise the **Introduction** and **Discussion** sections to further emphasize the points outlined above.
> > >
> > > Thank you again for your thoughtful feedback and valuable suggestions.

---

> > > > ### Author Response · Authors · 2024-12-03
> > > >
> > > > Thank you again for your insightful comments, which have been vital in refining our manuscript. As the discussion period concludes, we kindly ask if our response and updates, including additional experiments, sufficiently address your concerns. We would greatly value it if you could re-evaluate our paper based on these improvements.  If there are any issues or uncertainties, we are more than willing to address them swiftly within the remaining time. Your thoughtful feedback and support have been immensely helpful, and we deeply appreciate your time and effort in reviewing our work.

---

### Official Review · Reviewer_Lgkn · 2024-10-24

**Soundness:** 2
**Presentation:** 2
**Contribution:** 3
**Rating:** 6
**Confidence:** 3

**Summary:**

The paper is concerned with understanding variability in perceptual judgments between different people. It introduces a method for generating controversial stimuli, i.e. stimuli that elicit different classifications from different neural networks, in order to generate a dataset of human behavioral responses with high variability (varMNIST). Different classifiers are then finetuned on individual subjects' responses in varMNIST. These subject-specific classifiers are then used to generate a new set of controversial stimuli tailored to differences between individual subjects.

**Strengths:**

The idea of generating a dataset with the goal of having high variability between human behavioral responses is compelling and the paper nicely combines controversial stimuli and diffusion-based image generation to achieve this. I am very fond of the general approach of analyzing inter-individual variability instead of simply using neural networks to predict average human behavior and I think this is a promising direction. Overall, I found the paper well written and easy to follow. It starts with a clear introduction and description of what it sets out to do and Figure 2 really helps to understand the general idea of the dataset generation.

**Weaknesses:**

Unfortunately, I found it hard to understand some of the specifics of the methods and analyses, leading to several points of confusion and remaining questions, which are listed below and under "Questions". This also affects the reproducibility of the results, which I think would be really difficult to do just from the description of the methods in the paper, especially without any code.

Furthermore, I am not convinced that the results presented in the paper actually deliver on some of its central claims, that the generated images "more closely resemble natural images" or that the authors successfully "modulated individual behavior, unveiling significant inter-individual differences in perceptual variability". Details on these points also follow below. I admit that some of this might be due to me misunderstanding details of the analyses performed in the paper, which the authors might be able to clarify.

1. At first, I was confused about the dataset construction, in particular about what is taken as "ground truth" for varMNIST. After reading Section 3 "Human experiment and dataset construction", I got the impression that varMNIST contains 4,741 images, each with a label. After looking at Fig. 5b and reading Appendix A.3 and A.4, I think I understood that varMNIST contains a data point per subject per image, resulting in  102,021 labels in total. This should be clarified in the main text.
2. The abstract states that varMNIST contains 242,900 images, while Appendix A.4 suggests that it contains 102,021 after additional filtering. It is not clear which number applies to the actual dataset.
3. The discussion claims that "by utilizing carefully designed controversial stimuli, we selectively modulated individual behavior". I assume that this is about the controversial stimuli that were designed based on the subject-finetuned models in Section 6. I agree that this kind of analysis is a promising idea to investigate inter-individual variability in behavior. However, if I understand correctly, the newly generated stimuli are not actually tested in behavior. The way it is currently written, the paragraph "Controversial stimuli between subjects exhibit typical patterns" is misleading. It reads as if it is about disagreements between subjects in the experiment, but it is actually based on disagreements between models (which is clarified in the caption for Fig. 7). This is problematic because the subject-finetuned models seem to have a very low accuracy for quite a few subjects (see Fig. 5b). This should be clarified in the main text and the limitations of the analysis should be discussed. Whether the controversial designed for the subject-finetuned networks actually agree with the perceptual judgments of those subjects is an empirical questions that could be tested with a follow-up experiment.
4. The discussion of related work in the Methods section "Diffusion model as a regularizer to introduce prior information" seems a bit imprecise. In particular, it states that "a common issue was that the generated images lacked sufficient naturalness and failed to significantly influence human perception". But taking a quick look at one of the cited references (Gaziv et al., 2024), it seems that the generated images, while not necessarily being natural, did significantly influence human perception. The same is true for some of the other cited studies.
5. The claim that the proposed method results in images that "more closely resemble natural images" compared to previous methods for controversial stimuli is doubtful. Previous methods (e.g. Gaviz et al., 2024), which the authors of the present paper criticize as unnatural, are applied to photographic images like ImageNet. The present study is concerned with MNIST-style handwritten digits. It is not clear how the results transfer to photorealistic domains and if the diffusion-based approach would also lead to more "natural" images in these cases.
6. A couple of points about data visualization, which sometimes make it hard to follow the arguments made about the data.
    a. If I am not mistaken, Fig. 5 presents the same data in three different ways (average accuracy bar plot, single-subject accuracy point plot and kernel density). This could be simplified with something like a rainclould plot (https://wellcomeopenresearch.org/articles/4-63), which contains a point plot, kernel density estimate, and the average in a single plot.
    b. The color scheme is confusing. In Fig. 5a, different shades of red and blue indicate different models, while in Fig. 5b, some of the same colors are used to indicate training schemes. In Fig. 5c, the same traning schemes are then represented by different colors. In Fig. 6, again the same colors are used to represent out-cluster, in-cluster and subject-level predictions, and in Fig. 7 they represent different datasets. My recommendation is to use different colormaps to represent different concepts / distinctions.

Minor points:
- Fig. 7b is not referenced in the text
- The paragraph "Controversial stimuli between subjects constructed by finetuned models. " states that controversial stimuli were generated "using the algorithm in Section 3.1" (l. 401-402): There is no Section 3.1, so which algorithm was actually used?

**Questions:**

- About the analysis of the performance of the finetuned models (Fig. 5)
    - Was the varMNIST dataset split into a training and test set? Or were the models evaluated on the same images and labels on which they were finetuned? In other words, how can we be sure that the models are not overfitting on the individual subject data?
    - Does the "subject-finetuned (varMNIST-subject)" evaluation mean that each model that was finetuned on an individual subject's data was then tested on the data of that subject?
- Figure 6b: for some of the subjects, the subject-finetuned model performs worse at predicting in-subject compared to in-cluster. What is the explanation for this?
- How was the parameter $\alpha$, which trades off the entropy or controversy loss with the MSE loss, set?
- "we found that experimental images with response times between 300 and 5000 ms and entropy values between 0.5 and 2.5 best met our experimental requirements" (l. 716-717). What exactly were the experimental requirements?

---

> ### Author Response · Authors · 2024-11-25
> **Response to Reviewer Lgkn**
>
> Thank you for your constructive comments. We have made major revisions to the manuscript to address the issues you raised. The modifications can be found in the **Global rebuttal**.
>
> We have optimized the structure of the paper and supplemented it with detailed experimental descriptions. Please refer to **Global rebuttal 3** for more information.
>
> `W1&2. Dataset Construction`
>
>
> In the current version of the manuscript, we collected behavioral data from **400 participants**, each completing **500 trials**, resulting in a total of **200,000 trials** across **20,000 stimuli**. During data preprocessing, 154 participants were excluded based on control trials (also referred to as sentinel trials in the paper), leaving data from **246 participants**, comprising **116,715 trials** and **19,943 valid stimuli** in **varMNIST** (**Sec. 3.3.1**).
> This means that each stimulus was seen by approximately six participants, providing roughly six labels per stimulus. Based on this labeling, we calculated the entropy of each stimulus and analyzed its variability, enabling a robust discussion of perceptual variability within the dataset.
> All relevant descriptions in the manuscript have been updated accordingly.
>
> Additionally, we have updated the data collection process to improve the quality of the dataset. Please refer to **Global rebuttal 4** for more details.
>
> `W3. Manipulate individual behavior`
>
> Thank you for your detailed comments. In response to the issues you raised, we have made the following updates and improvements:
>
> We introduced new lab based behavioral experiments (see **Global rebuttal 1**) to further validate our generative model. These experiments demonstrate that the stimuli generated by our model can elicit significant controversial behavior between participants (**e.g., Participant A selects digit 4 while Participant B selects digit 5 on the same stimulus**).
> Notably, each participant labeled only about 500 samples in the first round of the experiment, yet their behavior was successfully manipulated in the second round. These results indicate that we were able to capture perceptual differences between individual participants even with a limited sample size.
>
> Regarding your concern that "subject-finetuned models seem to have very low accuracy for quite a few subjects," we have resolved this issue through updated experiments and fine-tuning methods. Among the 246 participants, the majority of individual-finetuned models achieved accuracies between 0.7 and 0.9, with only two participants falling below 0.6 (see **Fig. 4a and Sec 4.2**).
>
>
> `W4&5. Naturalness of Images`
>
> Our emphasis on natural images stems from our experimental goal: identifying stimuli that evoke **inter-participant variability**. For example, presenting a noise image to participants and asking them to choose a digit from 0 to 9 may lead to different choices each time, reflecting **intra-participant variability**. In contrast, we aim to find stimuli (if they exist) that are clearly reasonable for each participant's choice but provoke differing decisions between participants. Such variability could indicate inter-participant differences rather than random intra-participant variability. While we lack a comprehensive understanding of which image features evoke inter-participant variability, reducing intra-participant variability is a logical first step. Generating more natural images becomes essential in this context (e.g., a digit resembling both "3" and "5" and looks like a natural handwritten digit, as shown in **Sec. 3.2**).
>
> Gaziv et al.’s method, which constructs stimuli by adding perturbations to existing photographic images, generates results that are at least more natural than their baselines (based on adversarial examples), enabling stronger influences on human cognition. Similarly, the “naturalness” discussed here primarily refers to controlling image categories in a manner more aligned with human cognition (rather than introducing noise-like artifacts, as in adversarial examples), rather than focusing on whether the images belong to photorealistic domains. From this perspective, their method is comparatively “less natural” than newer adversarial example approaches based on diffusion models. To avoid misunderstanding, we have revised the description of Gaziv et al.’s work in the **Related Works** section.
>
> `W6. Visualization`
>
> Thank you for your suggestion. We have standardized the drawing style and labeling throughout the manuscript and modified the data visualization diagrams based on your feedback.
>
> `"W7. Controversial stimuli..."`
>
> We used the controversial guidance method described in **Sec. 3.1** of the updated paper.

---

> > ### Author Response · Authors · 2024-11-25
> > **Response to Reviewer Lgkn (pt. 2)**
> >
> > `Q1. Dataset split`
> >
> > The varMNIST dataset was split into a training set and a test set (8:2), with no overlap between the two. For group-level fine-tuning, the test set was composed of the test sets of all individual participants during individual fine-tuning. In other words, the test set of any individual participant was not included in the training set for either the group-level or individual-level fine-tuning. This approach ensures that the models do not overfit to the data from individual subjects (see**Sec. 4.1**).
> >
> > `Q2. "...tested on the data of that subject"`
> >
> > Yes, it does.
> >
> > `Q3. Some subject-finetuned model performs worse in-cluster`
> >
> > This phenomenon was primarily caused by the poor quality of the image dataset used in the previous experiment, which included many images that did not resemble digits, affecting the model's performance. In the new experiment, we addressed this issue by re-generating the dataset. An analysis of all 246 participants showed that individual fine-tuning improved prediction accuracy for 241 participants, with only 5 participants experiencing a slight decrease (Fig.4a).
> >
> > `Q4. "How was the parameter "\( \alpha \)...`
> >
> > For the generation process guided by multiple loss functions simultaneously, we employed the following approach: we checked post-hoc whether each loss function met the required thresholds. If not, the sample was regenerated. In practice, we used an empirical value of \( \alpha = 50 \). The primary purpose of using the MSE was to enhance the diversity of the generated samples. In the updated manuscript, we adopted a more efficient method to ensure diversity. Please refer to **Sec. 3.2** for details.
> >
> >
> > `Q5. "...What exactly were the experimental requirements?`
> >
> > The experimental requirements refer to our aim for the images to resemble digits while also eliciting perceptual variability in humans. This necessitated images with moderate response times (balanced difficulty) and entropy values (variability in human responses). However, in the new experiment, we removed this filtering step. This was for two main reasons: first, we optimized the generation process by introducing the digit judgment surrogate, ensuring that the generated images resemble digits; second, we recognized that samples with low human entropy are also meaningful, as they reflect differences in perceptual variability between humans and ANNs.

---

> ### Comment · Reviewer_Lgkn · 2024-11-28
>
> The authors have spent considerable effort on improving the weaknesses mentioned in the review. They have rerun their experiments with an improved focus on data quality, performed an additional experiment to manipulate individual behavior and improved the presentation of their results. I have therefore raised my score.
>
> W1&2. Dataset construction
> My confusion about the dataset construction and the number of data points has been resolved.
>
> My remaining concern is about the data filtering procedure. The data exclusion criterion resulted in 38.5% participants being excluded because they performed with less than 70% percent accuracy on standard MNIST images.
> - How was this criterion for exclusion determined?
>     - The proportion of participants that were excluded seems surprisingly high. Why did so many participants fail to classify simple handwritten digits? Recent work (e.g. Thomas et al., 2024) shows that dataset bias between large-scale online experiments and laboratory studies can affect the conclusions drawn from neural network models on online studies. I think this point warrants further discussion.
>
> W3. Manipulate individual behavior
>
> My concern that "subject-finetuned models seem to have very low accuracy for quite a few subjects" seems to be resolved, although it is not really clear to me which of the concrete steps taken by the authors ("novel directional sampling method", "digit judgment experiment", "updated experiments and fine-tuning methods") are responsible for this. Is it the due to the tweaks in the image generation and data collection or due to the substantial amount of data filtering (38.5% of participants)?
>
> I really appreciate that the authors actually performed the kind of experiment I suggested. The effect of controversial guidance on behavior, although relatively small, is statistically significant and provides a more direct test of the proposed method.
>
> W4&5. Naturalness of images
>
> Thanks for the explanation, I now understand what you mean. Like some of the other reviewers, I still have reservations about the naturalness of the images and the validity of the conclusions drawn from artificially generated controversial digits.
>
> W6. Visualization
>
> Thank you for the greatly improved figures with a consistent color scheme.
>
> Minor points:
> - "As shown in Figure. 5 and ??" (l. 431)
>
> Additional references:
> Thomas, T., Straub, D., Tatai, F., Shene, M., Tosik, T., Kersting, K., & Rothkopf, C. A. (2024). Modelling dataset bias in machine-learned theories of economic decision-making. Nature Human Behaviour, 8(4), 679-691.

---

> > ### Author Response · Authors · 2024-11-28
> >
> > ## Reply to Reviewer Lgkn
> >
> > Thank you for your detailed and thoughtful feedback. We are pleased that you recognize the improvements in our work. Below are our responses to your comments:
> >
> > ### W1&2. Dataset Construction
> >
> > We agree that the differences between online and laboratory experiments are becoming an increasingly important topic, especially as online experiments grow in prevalence. A key challenge with online experiments is the inability to effectively supervise whether participants are fully engaged. In our study, the average response time was ~1.5 seconds per trial, amounting to 10–15 minutes for 500 trials. During this time, some participants may lose patience and provide random answers to finish quickly and receive compensation. This behavior leads to a higher exclusion rate.
> >
> > Since our study lacked definitive “correct” answers for the main task, it was difficult to determine participants’ engagement levels. Thus, we randomly incorporated standard MNIST images into the trials to assess participant quality. We used a 70% accuracy threshold as our criterion, as shown in the table below. This threshold clearly separates participants into two distinct groups:
> >
> > | Accuracy | Count | Proportion |
> > |----------|-------|------------|
> > | 0.0      | 26    | 0.0650     |
> > | 0.1      | 44    | 0.1100     |
> > | 0.2      | 34    | 0.0850     |
> > | 0.3      | 20    | 0.0500     |
> > | 0.4      | 19    | 0.0475     |
> > | 0.5      | 6     | 0.0150     |
> > | 0.6      | 5     | 0.0125     |
> > | 0.7      | 11    | 0.0275     |
> > | 0.8      | 26    | 0.0650     |
> > | 0.9      | 72    | 0.1800     |
> > | 1.0      | 137   | 0.3425     |
> >
> > We will update the manuscript if accepted.
> >
> > ---
> >
> > ### W3. Manipulating Individual Behavior
> >
> > Regarding the improvement in individual models, we believe that all the reasons you mentioned contributed to the observed enhancement:
> >
> > 1. **Improved Data Generation Methods**: Our new approach, which uses guidance with target and a digit judgment surrogate, significantly improved data quality by avoiding overly difficult or ambiguous samples.
> > 2. **Participant Filtering**: The excluded participants likely disengaged midway or provided unreliable data, making their behavior difficult for models to predict. This is supported by the 38.5% exclusion rate, which appears to roughly align with the proportion of participants for whom individual models showed poor performance.
> >
> > ---
> >
> > ### W4&5. Naturalness of Images
> >
> > As we mentioned in our response to Reviewer SqUF, individual behavioral experiments are highly sensitive to task difficulty, particularly when the number of trials is limited. Complex stimuli often require more trials to learn participants’ preferences effectively. While we agree that natural images are an important direction for future work, we expect to explore this area further in future studies.
> >
> > ---
> >
> > ### Minor Points
> >
> > Thank you for pointing out the issue. This has been corrected.
> >
> > ---
> >
> > We sincerely appreciate your constructive comments again.
> >
> > Best wishes.
> >
> > All authors

---

> > > ### Author Response · Authors · 2024-12-02
> > >
> > > We greatly appreciate insightful suggestions from Reviewers **vTKk, Lgkn, and SqUF** regarding the use of **natural image stimuli**. Based on your feedback, we have conducted an additional experiment to address this concern.
> > >
> > > The new experiment includes all steps except manipulation, such as stimulus generation, data collection, and individual alignment. We generated 879 high-variability natural image stimuli and conducted a behavioral experiment with 30 participants. Due to the smaller dataset and the use of an MLP-head (instead of fine-tuning the entire model), the alignment effect is somewhat reduced compared on varMNIST. However, **both the behavioral experiments and individual alignment results confirm that the conclusions drawn on natural images are consistent with those on varMNIST**.
> > >
> > > The detailed results and figures can be found in **Global Rebuttal: Manuscript Update: Experiment on Natural Images**.
> > >
> > > We hope this update addresses your concerns and thank you for helping us improve the manuscript.

---

> > > > ### Author Response · Authors · 2024-12-03
> > > >
> > > > Dear Reviewer Lgkn, thank you once again for your detailed and thoughtful feedback, which has greatly helped us enhance our manuscript. As the discussion period is nearing its end, we would like to kindly ask if our response and the newly added experiments address your concerns. We would deeply appreciate it if you could re-evaluate our paper in light of these updates.
> > > > Should you have any additional questions or comments, please do not hesitate to let us know. We are committed to addressing any remaining uncertainties promptly. Your constructive insights and support have been invaluable throughout this process, and we are truly grateful for your time and effort.

---

### Official Review · Reviewer_SqUF · 2024-11-03

**Soundness:** 2
**Presentation:** 2
**Contribution:** 2
**Rating:** 6
**Confidence:** 3

**Summary:**

This paper presents a novel approach to studying human perceptual variability using artificial neural networks (ANNs). The authors develop a generative model that samples from the perceptual boundaries of ANNs to create synthetic MNIST images, inducing high variability in human perception. This method facilitates the construction of the varMNIST dataset, which is used for behavioral experiments involving 346 participants and over 240,000 trials. The research highlights that aligning the perceptual variability between ANNs and humans can predict human decision-making.

**Strengths:**

This paper presents an innovative approach to studying human perceptual variability by generating synthetic MNIST images from ANN perceptual boundaries, which are useful for extensive human behavioral experiments.

**Weaknesses:**

1. Scope of the dataset: The proposed method is only evaluated on MNIST, which limits the generalizability of the findings to real-world applications. While MNIST is effective for the initial exploration of perceptual variability, it does not capture the complexity inherent in more diverse visual datasets. This restricts the study's ability to address more nuanced or complex perceptual phenomena that may arise in real-world visual recognition tasks. For instance, datasets that include natural scenes, complex objects, or varied lighting and perspectives could provide more robust insights into human perceptual variability and ANN alignment.
2. Lack of quantitative comparison to baseline methods: Although the paper discusses the limitations of baseline methods in the related work section, it does not provide a quantitative comparison to those baseline approaches. This omission makes it difficult to objectively assess how much better sampling from the perceptual boundaries of ANNs performs in terms of aligning with human perceptual variability. A quantitative comparison would strengthen the paper's claims by demonstrating measurable improvements over baseline methods.
3. Trade-off between alignment and performance: While the paper shows that fine-tuning ANNs with human behavioral data improves perceptual alignment, there is a trade-off in performance on standard classification tasks, as pointed out by the authors. This suggests that optimizing models for perceptual alignment with human behavior may come at the cost of reducing their effectiveness for standard tasks.

**Questions:**

See Weaknesses

---

> ### Author Response · Authors · 2024-11-25
> **Response to Reviewer SqUF**
>
> Thank you for your valuable and insightful comments. We sincerely appreciate the time and effort you invested in reviewing our work. We have carefully considered your feedback and made significant revisions to the manuscript to address the issues you raised. Please refer to the **Global rebuttal** for detailed updates.
>
> `W1. Limitation the generalizability`
>
> Regarding the your concern about the focus on digit-like images rather than natural images, we acknowledge this limitation, which we also noted in the manuscript. While natural images are undoubtedly more relevant for many human perception studies, we chose to focus on digit recognition tasks to address a unique challenge: **evoking perceptual variability in a domain where participants typically share consistent views**. This makes digit recognition an effective and controlled test case for validating our methodolog, while natural images often don't elicit such shared views across observers. Also in general, given the novelity of this research activities it seems sensible to start a simple challenge before moving on to more difficult tasks. We hope the reviewer agrees with this research strategy.
>
>
> `W2. Lack of quantitative comparison`
>
> We sincerely thank your valuable suggestion. To enhance the reliability of our conclusions, we have supplemented our work with extensive experiments and analyses.
> Since systematic studies on human perceptual variability are still scarce, there are currently no existing methods that can be directly compared to ours (including those mentioned in the related works section). However, our behavioral analysis of varMNIST demonstrates that our generated dataset exhibits high entropy and guidance success rates, strongly indicating its ability to evoke human perceptual variability (see **Figure 3** in the revised manuscript).
> Additionally, we conducted extensive experiments and analyses to compare human and model performance on perceptual variability. These results confirm that our models can effectively predict human perceptual variability. For further details, please refer to **Global rebuttal 2**.
>
> Moreover, we introduced new behavioral experiments (see **Global rebuttal 1**) to further validate our generative model. These experiments demonstrate that the stimuli generated by our model can elicit significant controversial behavior between participants (**e.g., ensuring Participant A selects digit 4 while Participant B selects digit 5 on the same stimulus**). This is the **first time** such precise cross-participant modulation has been achieved in visual perception tasks. We believe this work will provide significant insights for future human behavior research and studies on AI-human alignment.
>
> `W3. Trade-off Between Alignment and Performance`
>
> The reviewer has raised an important point regarding the potential trade-off between perceptual alignment and performance on standard classification tasks. While this is a generally valid concern, our updated fine-tuned models (both group-level and individual-level) have demonstrated a surprising result: we **improved perceptual variability alignment** (i.e., prediction accuracy on varMNIST) while **maintaining performance on standard classification tasks** (i.e., prediction accuracy on MNIST)** (see **Fig. 4b, A.13**).
> A reasonable explanation for this phenomenon is that, for most participants, their predictions on the standard MNIST dataset are highly consistent. Therefore, even when the models learn participant-specific behaviors, their performance on MNIST remains unaffected.
>
> Upon further analysis, we found that individual and group-level models tend to differ primarily on more challenging stimuli (those with high entropy levels) while showing similar performance on simpler stimuli (see **Fig. 4d**). Moreover, the difference between individual-level and group-level models becomes more pronounced as task difficulty increases, underscoring the importance of individual alignment. This finding highlights the significance of human perceptual variability in AI-human alignment studies and suggests that studying such variability has important implications for advancing personalized AI systems.

---

> > ### Comment · Reviewer_SqUF · 2024-11-26
> > **Thank you for the discussion**
> >
> > Thank you to the authors for engaging in the discussion. I appreciate the time that put in to addressing my concerns and adding additional results. I raised my score. However, my primary concern was still the limited dataset used in the experiment, which significantly limits the generalizability of the claims in this paper. This primary concern was not sufficiently addressed. I think using natural images in the experiments will significantly improve the paper.

---

> > > ### Author Response · Authors · 2024-11-28
> > >
> > > We appreciate your feedback and agree that using natural images is an important direction for future work. Below, we outline the reasons for selecting MNIST in this study:
> > >
> > > 1. **Balancing Task Complexity and Experimental Precision**
> > >    There is a trade-off between task complexity and experimental precision. While MNIST digit recognition tasks may appear simplistic, they represent a step up in complexity from traditional perceptual decision-making studies, such as those involving basic stimuli like colors or angles. Moreover, MNIST provides a well-controlled environment for studying perceptual variability since each image contains only one clear object (a digit) without background distractions. In contrast, natural images introduce complexities like foreground-background separation and scene compositionality, which add many uncontrolled variables to data generation and behavioral experiments.
> > >
> > > 2. **Feasibility of Individual Behavioral Experiments**
> > >    Compared to group-level behavioral experiments, individual behavioral experiments often rely on limited data. Using simple datasets for such experiments is common practice in this field. For example, Gaziv et al. [2] used Restricted ImageNet with only 9 classes to guide participants' choices to specific categories. Similarly, Golan et al. [1] conducted human experiments on controversial stimuli generated from MNIST and CIFAR-10. In our study, focusing on individual preferences means that conducting experiments with more complex natural images would require substantially larger datasets, increasing experimental difficulty and resource demands. Hence, starting with MNIST allows us to effectively validate our methodology before scaling up to more complex datasets.
> > >
> > > 3. **Future Directions**
> > >    We fully acknowledge the value of using more diverse image datasets, and this is an area we are actively exploring. Another major direction is to expand the scope of visual recognition tasks to broader cognitive science applications, such as **similarity judgments**, **emotion recognition**, **visual attention**, and **scene memory**. These potential directions are discussed in greater detail in the updated **Discussion** section.
> > >
> > > References:
> > > 1. Golan, T., Raju, P. C., & Kriegeskorte, N. (2020). *Controversial stimuli: Pitting neural networks against each other as models of human cognition*. *PNAS*.
> > > 2. Gaziv, G., Lee, M. J., & DiCarlo, J. J. (2023). *Strong and Precise Modulation of Human Percepts via Robustified ANNs*. *NeurIPS*.

---

> > > ### Author Response · Authors · 2024-12-02
> > >
> > > We greatly appreciate insightful suggestions from Reviewers **vTKk, Lgkn, and SqUF** regarding the use of **natural image stimuli**. Based on your feedback, we have conducted an additional experiment to address this concern.
> > >
> > > The new experiment includes all steps except manipulation, such as stimulus generation, data collection, and individual alignment. We generated 879 high-variability natural image stimuli and conducted a behavioral experiment with 30 participants. Due to the smaller dataset and the use of an MLP-head (instead of fine-tuning the entire model), the alignment effect is somewhat reduced compared on varMNIST. However, **both the behavioral experiments and individual alignment results confirm that the conclusions drawn on natural images are consistent with those on varMNIST**.
> > >
> > > The detailed results and figures can be found in **Global Rebuttal: Manuscript Update: Experiment on Natural Images**.
> > >
> > > We hope this update addresses your concerns and thank you for helping us improve the manuscript.

---

> > > > ### Author Response · Authors · 2024-12-03
> > > >
> > > > Dear Reviewer SqUF, thank you again for your insightful feedback, which has been instrumental in improving our manuscript. As the discussion period is coming to a close, we kindly ask if our response and additional experiments address your concerns. We would be most grateful if you could re-evaluate our paper based on our updates. If you have any remaining questions or uncertainties, please let us know, and we will do our best to address them promptly within the remaining time. Your continued support and constructive feedback are invaluable to us, and we sincerely appreciate your time and effort.

---

> > > > > ### Comment · Reviewer_SqUF · 2024-12-03
> > > > > **Thank you for the additional experiments**
> > > > >
> > > > > Thank you for the response to my comments! My primary concern regarding using the MNIST dataset has been addressed with additional experiments on natural images. I appreciate the efforts the authors put into improving the quality of the paper, therefore I raised my score.

---

### Official Review · Reviewer_vTKk · 2024-11-04

**Soundness:** 2
**Presentation:** 2
**Contribution:** 2
**Rating:** 6
**Confidence:** 4

**Summary:**

The aim of this paper is to generate ambiguous digit-like images to study perceptual processing in humans. To this end, the authors propose a diffusion-based method for generating counterfactual digit-like images that produce conflicting predictions from humans; the diffusion process is guided by two kinds of signals, termed the uncertainty signal and the controversial signal. The uncertainty guidance encourages the diffusion model to produce images for which an AI visual classifier is confused between two labels for the images, while the controversial guidance encourages generation of images where two different classifiers produce diverging labels for each image. Further, the diffusion is guided by a mean-squared error loss to encourage generated images to be like real handwritten digits. This method is shown to be effective, as the generated images are stylistically similar to real images, but are confusing for humans to classify. Behavioral experiments conducted by the authors corroborate these claims. The authors term these generated images, together with the human classification data, the varMNIST dataset. Analysis of human prediction patterns for images from the varMNIST dataset shows slight positive correlation between an AI classifier’s prediction entropy and the entropy of human predictions on varMNIST images. By training AI classifiers using the human predictions, the authors showed that the varMNIST dataset can be used to identify controversial stimuli which reveal patterns of individual differences between subjects in predicting such ambiguous images.

**Strengths:**

The paper is well-motivated by existing work in cognitive science looking to study human perception with ambiguous stimuli. It also follows a recent body of work using AI models to study human perception. In this regard, the paper is tackling an interesting and important problem. There are several strengths of this paper, the most important of which are described below:

* The synthetic visual stimuli generated by the diffusion model using uncertainty and controversial guidance are of high quality — examples in the paper are stylistically similar to real digit-like images while evoking confusion among human subjects. Further, there seems to be significant variation in the degree of confusion (measured by entropy of human predictions) generated by the images. This kind of variability is highly desirable for such datasets which are intended for cognitive science research into human perception.
* Analysis in the paper shows that there is some degree of correlation between the entropy of visual AI classifiers and human classifiers on the images from the varMNIST dataset. This shows that the (uncertainty or controversial)-guided diffusion process generates synthetic images which capture some aspects of human perceptual boundaries.
* The authors’ use of the varMNIST dataset and the predictions from subjects to generate controversial stimuli that uncover individual differences amongst subjects is fascinating.
* The authors do a good job situating the paper in the context of existing work. The Introduction and Related Work sections do a good job putting this work in the context of existing research, while the Discussion section does a good job describing the contributions, implications and limitations of the work.

**Weaknesses:**

* The paper only considers the generation of ambiguous digit-like images, whereas for the purpose of human perception, natural images are much more relevant. In fairness, this weakness is acknowledged by the authors; nevertheless it remains a weakness.
* Important implementation details, such as the network and architecture details for the image generation, are missing in the paper. For the diffusion process itself, the appendix states that the authors used a pre-trained diffusion model. However, the architecture and the pretraining dataset are not described. Further, for the guidance signals, the architectures of the classifiers $f_1$ and $f_2$ have not been discussed. This significantly hampers both the reproducibility of the paper and the soundness of the work.
* Behavioral experiments require carefully designed checks to ensure that the dataset is balanced and not biased towards specific stimuli. Further, statistical analysis must be done after data collection to ensure that no bias inadvertently makes it into the dataset. The paper is lacking on both those counts. Some of these concerns are listed below:
    * The distribution over the 10 digits to be reference images in the final 4741 images is not presented. How many of the 4741 were generated using ’0’ or ‘1’ as the reference image? How many of these images were generated by uncertainty guidance and how many by controversial guidance?
    * The authors claim that the generated images have a random distribution over the range of entropies. This claim however is not supported by statistical tests. The histogram can be misleading since they are notoriously sensitive to the choice of parameters like the bin size.
    * In the random trials, how are the images to be displayed to subjects selected? Were there choices made to ensure that a subject sees the same number of synthetic images generated by using ‘0’ as the reference digit as the number of synthetic images generated by using ‘1’? Any kind of accidental subject-level bias can strongly affect patterns of individual differences studied later in the paper.

* The description for the finetuning process was quite unclear.
    * Were the labels of the reference images used as the correct labels to calculate the accuracy on varMNIST? I’m not sure about such a choice because the motivation for the dataset is to generate ambiguous images and having fixed labels for such images seems counterintuitive.
    * What are the targets used to finetune the models when using population-level data? Is it the average over predictions for a particular image over all subjects?

* The writing in the paper can be improved to improve clarity and readability of the paper.
    * One of the main contributions of the paper is the varMNIST dataset and the diffusion process used to generate the images. It is surprising that a lot of highly relevant details regarding the image generation are included only in the appendix. I would strongly recommend the authors to update the manuscript to include pertinent details, at least briefly, in the main paper.
    * The same point holds about the behavioral experiments. Details such as the number of subjects, number of trials and rationale for choosing the criteria for selecting the 4741 images are better placed in the main paper because they are central to the paper’s contributions.

* A couple of minor typos:
    * perveptual -> perceptual in section 5 title (line 310)
    * There is no section 3.1 (line 402)

**Questions:**

I have several questions which I would like the authors to respond to. I have stated them in the 'Weaknesses' section of the review, but I will list them here for the authors' convenience:
* How many of the 4741 were generated using ’0’ or ‘1’ as the reference image? How many of these images were generated by uncertainty guidance and how many by controversial guidance?
* How statistically significant are the claims of randomness of the distribution of human prediction accuracy?
* How are images to be displayed to subjects selected? What design choices were made to ensure that subjects were equally likely to view images generated by different reference images?
* What is the correct label for an image from the varMNIST dataset? What was the rationale for the choice you made?
* In finetuning the models, what are the targets used in the case of population-level alignment?

In addition, I have a couple of additional questions for which I would appreciate the authors' clarifications:
* Why is the MSE loss important? The authors claim that bias in the prior distribution of MNIST causes generated images to be biased towards specific digits. Would this not be solved by sampling from MNIST so that all digits are equally likely to be selected as reference images?
* Why is model entropy (Fig 4a) centered around 0?
* Will the varMNIST dataset along with the results of the behavioral experiments be made publicly available?

---

> ### Author Response · Authors · 2024-11-25
> **Response to Reviewer vTKk**
>
> We sincerely thank your detailed evaluation of our work. We have made major revisions to the manuscript to address the issues you raised. The modifications can be found in the **Global rebuttal**.
>
> `W1. Natural images as stimuli`
>
> Regarding the your concern about the focus on digit-like images rather than natural images, we acknowledge this limitation, which we also noted in the manuscript. While natural images are undoubtedly more relevant for many human perception studies, we chose to focus on digit recognition tasks to address a unique challenge: **evoking perceptual variability in a domain where participants typically share consistent views**. This makes digit recognition an effective and controlled test case for validating our methodology, while natural images often don't elicit such shared views across observers. Also in general, given the novelity of this research activities it seems sensible to start a simple challenge before moving on to more difficult tasks. We hope the reviewer agrees with this research strategy.
>
> `W2. Missing Implementation Details`
>
> We have updated the manuscript and the relevant appendix to ensure all technical details are clear and complete. Please refer to **Global rebuttal 3** for more information.
>
> - The diffusion model is not a pre-trained model but was trained by us. Thank you for pointing out this error, which we have now corrected. Details of the diffusion model have been added to **Appendix A.3.1**.
> - The architectures of the classifiers are described in **Appendix A.3.2**. The performance of different classifiers in generating high variability images is thoroughly discussed in **Fig. 3b, A.11**.
>
> `W3. Balance of dataset`
>
> Your concern is highly relevant, and we have addressed it by improving the data generation algorithm and experimental process, followed by re-collecting the data. Please refer to **Global rebuttal 4**. We believe the issue you raised has been effectively resolved.
>
> - In the original paper, reference digits were uniformly sampled from the standard MNIST dataset. The images used as experimental stimuli (20,000 in total) are detailed in **Table 3** (distribution of different generation strategies), **Fig. A.6** (distribution of guided digits). **Sec. A.2**, and **Fig. A.3, A.4, and A.5** further analyze the effects of different methods on stimulus distribution and diversity.
>
> - In the updated paper, our method ensures high uncertainty in the model's predictions for generated images (**Appendix A.2.1**). Regarding the entropy of human decisions, additional discussions can be found in **Fig. A.10 (bottom left)** and **Fig. 4c, d**.
>
> - As detailed in **Appendix A.2.1, A2.4** and **Fig. A.6**,  the images viewed by each participant were sampled evenly from the generated dataset of 20,000 images.
>
>
> `W4. Details for finetuning`
>
> The details of the finetuning process have been supplemented in **Sec. 4.1** and **Appendix B.2**.
>
> - During the finetuning process, only participant choices were used as labels. For our task, there is no "correct" label, as different participants may make different choices for the same stimulus. This is why the individual model (IndivNet in the paper) is meaningful and necessary.
> - The group model (GroupNet in the paper) was trained on the entire varMNIST dataset, which includes all participant choices. During training, the model encounters cases where the same stimulus is associated with multiple labels. However, the model's predictions ultimately align with the majority opinion (i.e., the choice made by most participants).
>
> `W5. Improve clarity and readability`
>
> See **Global rebuttal: 3**.
>
> `Q1. Why is the MSE loss important?`
>
> In the original paper, MSE loss was used to mitigate the bias of the diffusion model, which tended to generate a simpler images due to the imbalance in the prior distribution. Since the guidance method did not specify target digits, MSE loss was necessary to guide generation toward specific digits, ensuring balanced sampling. Reference digits were directly sampled from the standard MNIST dataset.
> In the revised paper, we addressed this issue more directly by specifying target digits for guidance (see **Fig. A.3, A.4, A.5**), eliminating the need for MSE loss.
>
> `Q2. Why is model entropy centered around 0?`
>
> This is because the classifiers trained on the MNIST dataset (base model) exhibits overfitting. When encountering out-of-distribution samples like varMNIST, the models tend to produce overconfident predictions, resulting in model entropy being centered around lower values.

---

> > ### Author Response · Authors · 2024-11-25
> > **Response to Reviewer vTKk (pt. 2)**
> >
> > `Q3. Publicly available`
> >
> > All data and code will be made publicly available. We are currently organizing the code and dataset and aim to release an initial version before the discussion stage concludes.

---

> > > ### Author Response · Authors · 2024-11-28
> > >
> > > We have updated and uploaded the code and dataset. They are now accessible at the following link:
> > > [https://anonymous.4open.science/r/HumanPerceptualVariability](https://anonymous.4open.science/r/HumanPerceptualVariability)
> > >
> > > Other updates can be found in the **Global Rebuttal**.

---

> > > > ### Author Response · Authors · 2024-12-02
> > > >
> > > > We greatly appreciate insightful suggestions from Reviewers **vTKk, Lgkn, and SqUF** regarding the use of **natural image stimuli**. Based on your feedback, we have conducted an additional experiment to address this concern.
> > > >
> > > > The new experiment includes all steps except manipulation, such as stimulus generation, data collection, and individual alignment. We generated 879 high-variability natural image stimuli and conducted a behavioral experiment with 30 participants. Due to the smaller dataset and the use of an MLP-head (instead of fine-tuning the entire model), the alignment effect is somewhat reduced compared on varMNIST. However, **both the behavioral experiments and individual alignment results confirm that the conclusions drawn on natural images are consistent with those on varMNIST**.
> > > >
> > > > The detailed results and figures can be found in **Global Rebuttal: Manuscript Update: Experiment on Natural Images**.
> > > >
> > > > We hope this update addresses your concerns and thank you for helping us improve the manuscript.

---

> > > > > ### Comment · Reviewer_vTKk · 2024-12-02
> > > > > **Official comment by reviewer vTKk**
> > > > >
> > > > > I appreciate the time and effort the authors put into improving the manuscript and answering the reviewers' questions. The additional analysis of the varMNIST dataset addressed my concerns about bias in the dataset. Similarly, the updated manuscript and visualizations address my concerns about missing experiment details and aid both reproducibility and comprehension. Therefore, I have raised my score.

---

### Author Response · Authors · 2024-11-25
**Global Response (pt. 1)**

# Global Rebuttal

We sincerely thank the reviewers for their thorough review and valuable comments on our paper. We have carefully reviewed and fully considered the suggestions and recommendations provided by the four reviewers. Based on this feedback, we have conducted additional experiments and analyses and made comprehensive revisions to the manuscript while preserving the original conclusions. Since we have updated the structure of the manuscript to better address the reviewers’ concerns, we have summarized the revisions into the following four key points instead of highlighting specific changes in the text.

### 1. Behavioural experiment of manipulating human perceptual variability. (Sec. 5 and Fig. 5)

Based on the concerns raised by reviewer Lgkn, we conducted a lab-based human behavioral experiment, building upon the original Section 6 (now revised as Section 5). The aim of the experiment was to validate whether the high-variability stimuli generated by our method can efficiently align with individual responses and whether the generative model can produce stimuli that elicit significant controversial behavior between two participants. This experiment would significantly enhances the validity of our paper.

Through directional, participant-specific precise manipulation (e.g., **Participant A selects digit *4* while Participant B selects digit *5* on the same stimulus**), we demonstrated that our model successfully captured the key factors driving differences between participants. Specifically, we conducted two rounds of experiments with 18 participants:
Round 1: Participants completed a digit recognition task with stimuli from the varMNIST dataset to determine initial behavioral data.
Round 2: Using the data collected in Round 1, we fine-tuned the models and generated a new set of experimental stimuli with the generative model.

Our results demonstrated that our method significantly **improved the targeted ratios across almost all participant pairs**. This indicates that, **when presented with the same stimulus, both participants were more inclined to choose the numbers aligned with their respective guidance targets**, further validating our model’s ability to decode and manipulate perceptual variability.
(**see Fig. 5**)


### 2. Deeper analysis of behavioral data and human-model alignment (Sec. 3.3.2 and Sec. 4, Fig. 3 & 4)

Based on the suggestions from reviewer vTKk, SqUF, Lgkn and vi4H, we performed a more in-depth and comprehensive analysis of the behavioral data and human-model alignment.

- **Quantitative analysis on varMNIST**: We conducted a quantitative analysis that includes 1) new evaluation metrics; 2) ANN variability can arouse human variability; 3) Guidance strategy, classifier and target influence the guidance outcome (**see Sec. 3.3.2, Fig. 3 and Fig. A.10, A.11, A.12**).

- **Alignment analysis**: We conducted a comprehensive analysis of the alignment between human behaviors and models in group and individual level. Specifically, we discussed: 1) Fine-tuning improves both group-level and individual-level prediction performance; 2) Different classifiers exhibit inconsistent performance; 3) Human variability can be predicted by models; 4) Performance of behavior prediction across images with varying entropy levels. (see **Sec. 4, Fig. 4 and Fig. A.13, A.14, A.15**)

### 3. New written content

Based on the suggestions from reviewer vTKk, Lgkn and vi4H, we made the following revisions:

1) **Article structure:** We reorganized the article into three sections: (**Section 3**) "Collecting human perceptual variability," (**Section 4**) "Predicting human perceptual variability," and (**Section 5**) "Manipulating human perceptual variability." Additionally, we provided a comprehensive overview of the content in the Introduction. We also updated **Fig. 1** and the para 3 of Introduction to better reflect the revised structure and clarified that our core contribution lies in proposing a systematic paradigm for studying human perceptual variability.
2) **Revised figures:** We redesigned figures, **including** methodological illustrations for each section (**Fig. 1 and Fig. 2**), and added several result and evaluation figures (**Fig 3, Fig 4b&d and Fig5 b&c**).
3) **Methods and experimental details:** The main methods and experimental content are detailed in the main text (**Sec. 3**), with additional details provided in the Appendix (**Appendix A 2&3**).

---

> ### Author Response · Authors · 2024-11-25
> **Global Response (pt. 2)**
>
> ### 4. Improved data collection process to ensure high-quality experimental stimuli (Sec. 3)
>
> Based on the suggestions from reviewer vTKk, Lgkn and vi4H, we improved our data collection process to ensure the acquisition of high-quality experimental stimuli. Unlike our orignal non-directional uncertainty guidance and controversial guidance methods (based on entropy and KL divergence), we introduced a novel directional sampling method (simultaneously guiding toward digits *i* and *j*) to sample more evenly from the ANN's decision boundaries. This resulted in a more diverse dataset (**Sec. 3.1 and Appendix A.2.1&A.2.4**).
>
> Additionally, to ensure sample quality and avoid generating stimuli that participants would find impossible to classify, we conducted a digit judgment experiment where humans were asked to "is this picture a digit?". The data from this experiment were used to train a judgment surrogate model, which was then integrated into our method for image generation. This step helped preventing out-of-distribution samples and improved the quality of stimuli in varMNIST (**Sec 3.2**).
>
> To further enhance the quality of the experimental data, we included control trials (also referred to as sentinel trials in paper) from the standard MNIST dataset. Only participants with an accuracy of 70% or higher on these control trials were included in the final dataset. This step eliminated 38.5% of participants. With this measure, we successfully constructed a higher-quality and more reliable varMNIST dataset (**Sec. 3.3**).

---

> > ### Author Response · Authors · 2024-11-27
> > **Manuscript Update**
> >
> > We have made the following updates to the manuscript:
> >
> > 1. **Expanded Results on Experiments of Manipulating Human Perceptual Variability**:
> >    - Figures A.17 to A.19 now present detailed experimental results for pairwise interactions between participants 1, 2, and 3, including all positive and negative examples of stimuli.
> >    - Figure A.20 provides an analysis of manipulation performance, highlighting prediction accuracy, guidance success rate, and target ratio for each subject pair.
> >
> > 2. **Updated Related Works**:
> >    - The section has been slightly revised to include updated references and discussions.
> >
> > We look forward to further discussions with the reviewers, which greatly helped us improve the manuscript.

---

> > > ### Author Response · Authors · 2024-11-28
> > > **Manuscript Update**
> > >
> > > We have made the following updates:
> > >
> > > 1. **Upload Code and Dataset**
> > >    - The code and dataset have been uploaded and are accessible at the following link:
> > >      https://anonymous.4open.science/r/HumanPerceptualVariability
> > >
> > > 2. **Improved Figure 5a**
> > >    - The left panel now showcases examples of success (positive, negative), bias, and failure cases, providing clearer illustrations of the choices made by individual models and human participants.
> > >
> > > 3. **Expanded Discussion**
> > >    - Added discussions on exploring individual variability using datasets with natural images and other cognitive tasks, highlighting potential directions for future research.
> > >
> > > We look forward to further discussions with the reviewers, which have greatly helped us improve the manuscript.

---

> ### Author Response · Authors · 2024-12-02
> **Manuscript Update: Experiment on Natural Images**
>
> We have conducted new experiments in response to suggestions from reviewers vTKk, Lgkn, and SqUF.
>
> ### **Experiment Details**
>
> Given the lack of prior experiments with natural images, conducting a full-scale study within one week posed significant challenges, particularly due to the complexity of cognitive experiments and subsequent analyses. Nevertheless, we performed a small-scale experiment with a limited number of participants, and obtained preliminary results, which are summarized below. These figures will soon be available at the following link:
> [https://anonymous.4open.science/r/HumanPerceptualVariability](https://anonymous.4open.science/r/HumanPerceptualVariability)
>
>
> ### **Key Findings**
>
> 1. **Evoking Perceptual Variability**
>    - Sampling along ANN boundaries successfully evoked perceptual differences in humans using natural image stimuli.
>    - Example stimuli generated by our method are shown in Fig. B.1, while behavioral results are presented in Fig. B.3.
>    - Participant responses (success, bias, and failure) were analyzed, similar to Fig. 3 in the manuscript. The analysis and corresponding data are shown in Fig. B.4 and the table below:
>
>    | Guidance Outcome | Proportion |
>    |------------------|------------|
>    | Success          | 0.571104   |
>    | Bias          | 0.376564   |
>    | Failed           | 0.052332   |
>
>    - Typical stimuli with the highest success rates are presented in Fig. B.2. Our findings indicate that stimuli generated from natural images can successfully influence participants' choices with high variability.
>
> 2. **Capturing Human Perceptual Differences**
>    - Even with fine-tuning on an MLP head of CLIP, we demonstrated the ability to **capture human perceptual differences**. Results are detailed in Fig. B.5 and summarized in the tables below:
>
>    **Performance on the Group-Level Validation Set**
>
>    | Model       | Accuracy  |
>    |-------------|-----------|
>    | Base model  | 0.571319  |
>    | GroupNet    | 0.614264  |
>
>    **Performance on the Individual-Level Validation Set  (noise ceiling is 66.48%)**
>
>    | Model       | Accuracy Mean | Accuracy Std  |
>    |-------------|---------------|---------------|
>    | GroupNet    | 0.613882      | 0.2327        |
>    | IndivNet    | 0.628444      | 0.24012       |
>
> ### **Experimental Setup**
>
> We followed the experimental paradigm of Gaziv et al. [1] and referenced methods from Chen et al. [2][3]. We introduced uncertainty guidance to generate stimuli using natural image datasets.
>
> - **Dataset**:
>    - Used 9 classes ('Dog', 'Cat', 'Frog', 'Turtle', 'Bird', 'Monkey', 'Fish', 'Crab', 'Insect') from Restricted ImageNet [1][4].
>    - Filtered synthetic data yielded 879 images.
>
> - **Participants**:
>    - Recruited 30 participants, each completing 300 trials.
>    - Excluded data from 8 participants based on response times, retaining data from 22 participants (6600 trials in total).
>
> - **Procedure**:
>    - A 200ms fixation cross was displayed before each stimulus, which was shown for 200ms.
>    - Participants made their choices without time constraints.
>
> ### References
>
> 1. Guy Gaziv, Michael Lee, and James J. DiCarlo. *Strong and Precise Modulation of Human Percepts via Robustified ANNs.* Advances in Neural Information Processing Systems, 36, 2024.
> 2. Chen Wei, Jiachen Zou, Dietmar Heinke, and Quanying Liu. *Cocog: Controllable Visual Stimuli Generation Based on Human Concept Representations.* IJCAI-24, 2024a.
> 3. Chen Wei, Jiachen Zou, Dietmar Heinke, and Quanying Liu. *Cocog-2: Controllable Generation of Visual Stimuli for Understanding Human Concept Representation.* arXiv:2407.14949, 2024b.
> 4. Logan Engstrom, Andrew Ilyas, et al. *Robustness (Python Library),* 2019.

---

> > ### Author Response · Authors · 2024-12-02
> > **Manuscript Update: Experiment on Natural Images (Pt. 2)**
> >
> > These include a new experiment exploring natural images and its corresponding figures.
> >
> > The figures and detailed results are now available at the following link:
> > [https://anonymous.4open.science/r/HumanPerceptualVariability/experiment_natural_images/figures.md](https://anonymous.4open.science/r/HumanPerceptualVariability/experiment_natural_images/figures.md)
> >
> > We hope this additional analysis addresses the reviewers' concerns and provides valuable insights. Thank you for your ongoing feedback, which has significantly enhanced our work.

---

### Meta-Review · Area_Chair_enGL · 2024-12-08

**Metareview:**

This submission introduces a methodology for generating stimuli that elicit varied perceptual responses across both neural network classifiers and human subjects, while also demonstrating the capacity to fine-tune neural networks to account for individual human perceptual differences. The original manuscript focused exclusively on the MNIST dataset for experimental validation. While the investigation of perceptual variability and its alignment between artificial and biological vision systems represents a valuable research direction, reviewers initially expressed concerns regarding the reliance on non-naturalistic MNIST data. The authors subsequently provided additional experiments using ImageNet, but these results were submitted after the response period had concluded. This late submission of significant new experimental results would necessitate another round of review for proper evaluation.

**Additional Comments On Reviewer Discussion:**

The authors made several key revisions during the official response period, including enhanced statistical analysis with significance testing, improved data quality measures through participant screening, and clearer explanation of their methodological framework. After the response period concluded, they conducted substantial new experiments using natural images from ImageNet. While these additional experiments may address reviewer concerns about dataset limitations, they represent a significant addition outside the formal review process that would require another round of evaluation with these results incorporated into the main text.

---

### Decision · Program_Chairs · 2025-01-22

Reject